# Genetically defined nucleus incertus neurons differ in connectivity and function

Emma D Spikol[1,2], Ji Cheng[1,3], Michelle Macurak[3], Abhignya Subedi[3], Marnie E Halpern[1,2,3]*

[1]Department of Molecular and Systems Biology, Geisel School of Medicine at Dartmouth, Hanover, United States; [2]Department of Neuroscience, Johns Hopkins University School of Medicine, Baltimore, United States; [3]Department of Biology, Johns Hopkins University, Baltimore, United States

*For correspondence:
Marnie.E.Halpern@Dartmouth.edu

Competing interest: The authors declare that no competing interests exist.

**Abstract** The nucleus incertus (NI), a conserved hindbrain structure implicated in the stress response, arousal, and memory, is a major site for production of the neuropeptide relaxin-3. On the basis of *goosecoid homeobox 2* (*gsc2*) expression, we identified a neuronal cluster that lies adjacent to *relaxin 3a* (*rln3a*) neurons in the zebrafish analogue of the NI. To delineate the characteristics of the *gsc2* and *rln3a* NI neurons, we used CRISPR/Cas9 targeted integration to drive gene expression specifically in each neuronal group, and found that they differ in their efferent and afferent connectivity, spontaneous activity, and functional properties. *gsc2* and *rln3a* NI neurons have widely divergent projection patterns and innervate distinct subregions of the midbrain interpeduncular nucleus (IPN). Whereas *gsc2* neurons are activated more robustly by electric shock, *rln3a* neurons exhibit spontaneous fluctuations in calcium signaling and regulate locomotor activity. Our findings define heterogeneous neurons in the NI and provide new tools to probe its diverse functions.

## eLife assessment

This study presents an **important** finding on the anatomical connectivity and functional roles of the previously uncharacterized neuronal populations in the nucleus incertus. The evidence supporting the conclusions is **convincing**, with imaging and manipulations of the genetically targeted populations of neurons. The work presents a significant milestone for future mechanistic studies of the nucleus incertus.

## Introduction

The nucleus incertus (NI), originally identified in the human brain (*Streeter, 1903*), consists of bilaterally paired clusters of neurons at the midline of the floor of the fourth ventricle (*Ma and Gundlach, 2015*; *Olucha-Bordonau et al., 2018*). A variety of neuropeptides have been detected in the region, including cholecystokinin (*Olucha-Bordonau et al., 2003*), neuromedin B (*Lu et al., 2020*), neurotensin (*Jennes et al., 1982*), and relaxin-3 (*Burazin et al., 2002*; *Smith et al., 2010*); however, the properties of NI neuronal subtypes are not well defined.

Initial investigations in rodents indicate that the NI responds to stressful cues; NI neurons are enriched in receptors for the Corticotropin Releasing Factor (CRF) and upregulate c-Fos in response to CRF exposure (*Bittencourt and Sawchenko, 2000*; *Potter et al., 1994*). Placement in an elevated plus maze, exposure to an anxiogenic drug, foot shock, or water-restraint stress also induce expression of the neural activity marker c-Fos in the NI (*Lawther et al., 2015*; *Passerin et al., 2000*; *Rajkumar*

*et al., 2016*). Other reports have implicated the NI in regulating baseline locomotor activity. For example, electrical microstimulation of the NI promotes locomotion in rats (*Farooq et al., 2016*), and optogenetic activation of a subset of neurons in the mouse NI that produce the neuropeptide neuromedin B increases locomotor speed (*Lu et al., 2020*).

In rodents, the NI contains the largest population of neurons in the brain that produce relaxin-3 (RLN3; *Smith et al., 2011*; *Tanaka et al., 2005*), a neuropeptide thought to mediate behavioral responses to aversive stimuli (*Lawther et al., 2015*; *Ryan et al., 2013*; *Zhang et al., 2015*). Although there are also NI neurons that do not produce RLN3 (*Ma et al., 2013*), their characteristics are not well distinguished from the RLN3 population.

Larval zebrafish are a powerful model to investigate neuronal diversity and connectivity because their transparency and genetic tractability are advantageous for monitoring and manipulating specific subpopulations. In zebrafish, the presumed analogue of the NI is the griseum centrale, situated on the ventral surface of the rhombencephalic ventricle (*Agetsuma et al., 2010*; *Olson et al., 2017*; *Wullimann et al., 1996*). Expression of *relaxin 3a* (rln3a) is restricted to two bilaterally paired clusters of neurons in the midbrain and two bilaterally paired nuclei bordering the hindbrain midline (*Donizetti et al., 2008*). It was proposed that the midbrain *rln3a* expression domains correspond to the periaqueductal gray (PAG), a region that produces RLN3 in rodents (*Ma et al., 2017*; *Smith et al., 2010*; *Tanaka et al., 2005*), and that the hindbrain *rln3a* neuronal clusters correspond to the NI (*Donizetti et al., 2008*).

The zebrafish griseum centrale is a proposed target of the habenulo-interpeduncular nucleus (Hb-IPN) axis, a highly conserved forebrain to midbrain pathway implicated in modulating anxiety and the response to aversive stimuli (*Agetsuma et al., 2010*; *Duboué et al., 2017*; *Facchin et al., 2015*; *McLaughlin et al., 2017*). Left-right asymmetry of the habenular region is widespread among vertebrate species (*Concha and Wilson, 2001*; *Harris et al., 1996*) and in zebrafish the left and right dorsal habenulae (LdHb and RdHb) exhibit prominent differences in their molecular properties, connectivity, and functions (*Agetsuma et al., 2010*; *Chou et al., 2016*; *deCarvalho et al., 2014*; *Dreosti et al., 2014*; *Duboué et al., 2017*; *Facchin et al., 2015*; *Gamse et al., 2005*). The LdHb projects to the dorsal IPN (dIPN) and ventral IPN (vIPN), whereas RdHb neurons largely innervate the vIPN (*Gamse et al., 2005*). Using tract tracing in adult zebrafish, *Agetsuma et al., 2010* found that vIPN neurons project to the dorsal raphe and dIPN neurons to the hindbrain griseum centrale. Moreover, injection of the cell-filling dye neurobiotin into the dorsal IPN resulted in labeling of cell bodies in the griseum centrale, suggesting reciprocal connectivity (*Agetsuma et al., 2010*). The NI and IPN are also reciprocally connected in rodents (*Goto et al., 2001*; *Olucha-Bordonau et al., 2003*). However, whether different neuronal populations in the hindbrain NI innervate distinct subregions of the IPN is unresolved.

In this study, we find that a small population of neurons defined by expression of the *goosecoid homeobox 2* (gsc2) gene is closely apposed to *rln3a* neurons in the zebrafish hindbrain and distinct from neurons producing relaxin-3, cholecystokinin, and neuromedin B. Through CRISPR/Cas9-mediated targeted integration, we generated transgenic driver lines to facilitate selective labeling and manipulation of the *gsc2* and *rln3a* neuronal populations in the nucleus incertus, and found that they differ in efferent and afferent connectivity, calcium signaling, and control of locomotor behavior.

## Results

### Identification of *gsc2* neurons in the nucleus incertus

We initially identified the *gsc2* gene through transcriptional profiling aimed at distinguishing genes with enriched expression in the midbrain interpeduncular nucleus (IPN). IPN tissue was microdissected from the brains of adult zebrafish harboring *TgBAC(gng8:Eco.NfsB-2A-CAAX-GFP)c375*, a transgene that labels dorsal habenular (dHb) neurons and their axons with membrane-targeted GFP in the larval and adult brain (*deCarvalho et al., 2013*). Because GFP-labeled dHb axon terminals demarcate the IPN, they serve as a guide to locate and excise this midbrain structure. After comparing the transcriptional profile of pooled IPN samples with remaining brain tissue, *gsc2* transcripts were identified as enriched approximately fivefold in the IPN region relative to the rest of the brain. The *gsc2* gene encodes a protein that has homology to Goosecoid-related proteins in its homeobox

domain-containing sequence. We note that the *gsc2* sequence is not annotated in the latest genome assembly (GRCz11) and was initially identified by aligning reads to Zv9 (Ensembl release 77).

From whole-mount in situ hybridization (WISH), we found that *gsc2* transcripts are restricted to bilateral clusters just posterior to the midbrain-hindbrain boundary and to a few sparsely distributed neurons anterior to the main cluster (**Figure 1A and A'**). Double labeling with *somatostatin 1.1* (*sst1.1*), a marker of IPN neurons (**Doll et al., 2011**), revealed that the bilateral clusters are not situated within the IPN but rather lie dorsal to it (**Figure 1B and B'**).

Owing to the similar positions of *gsc2* and *rln3a* (**Donizetti et al., 2008**) neurons in the larval hindbrain, we performed double-label WISH, and found that *gsc2* neurons are a distinct population, located anterior to the *rln3a* neurons (**Figure 1C, C' and D**).

Other neuropeptides in addition to RLN3 have been detected in the rodent NI, including neuromedin B in mice (**Lu et al., 2020**), and cholecystokinin (**Kubota et al., 1983**; **Olucha-Bordonau et al., 2003**) and neurotensin (**Jennes et al., 1982**) in rats. To determine whether transcripts encoding each of these neuropeptides are present in the zebrafish NI, we performed WISH for the zebrafish *cholecystokinin a* (*ccka*), *cholecystokinin b* (*cckb*), *neuromedin a* (*nmba*), *neuromedin b* (*nmbb*), and *neurotensin* (*nts*) genes (**Figure 1—figure supplement 1A–E'**). For *cholecystokinin* and *neuromedin B*, the combined expression of the two zebrafish paralogues closely resembles the overall expression pattern of each single rodent gene (**Albus, 1988**; **Ohki-Hamazaki, 2000**). Only *cckb* and *nmbb* transcripts were detected in the NI, and *nmbb* expression was also observed in the PAG (**Figure 1—figure supplement 1B**, D). Using double-label fluorescent WISH, we found that the *gsc2* neurons (48.33±2.33 neurons) fail to express any of these neuropeptides and comprise a unique population. We found that hindbrain *nmbb* neurons (8.33±1.45) are intermingled with *rln3a* neurons (10.67±1.33) in the NI, with a small subset expressing both neuropeptides (**Figure 1E**, **Figure 1—figure supplement 2A'-C''**). By contrast, *rln3a* and *nmbb* neurons exist as separate, adjacent populations in the PAG (**Figure 1—figure supplement 2A**). Hindbrain *cckb* neurons (4.5±1.1) are a distinct population located just posterior to *rln3a* and *nmbb* neurons (**Figure 1F**). From these results, we can construct a map of peptidergic neurons in the zebrafish NI (**Figure 1G**), with a discrete group of *gsc2*-expressing neurons, partially overlapping expression of *rln3a* and *nmbb* in cells posterior to the *gsc2* neurons, and a distinct population of *cckb* neurons posterior to the *rln3a* and *nmbb* neurons.

## *gsc2* and *rln3a* transgenic lines drive expression in the NI

To verify that the *gsc2* and *rln3a* neurons reside in the zebrafish analogue of the mammalian NI, we examined the properties of these closely apposed neuronal populations. Using CRISPR/Cas9-mediated genome integration, we generated transgenic lines to selectively label and manipulate each group. The *gsc2* and *rln3a* loci were independently targeted for integration of sequences encoding QF2 (**Figure 2A and D**), a modified transcription factor that binds to the upstream activating sequence (QUAS) in the bipartite Q transcriptional regulatory system of *Neurospora crassa* (**Riabinina and Potter, 2016**; **Subedi et al., 2014**). *Tg(gsc2:QF2)^c721^* was generated by introducing the *QF2* sequence into exon 2 of the *gsc2* gene through non-homologous end joining (**Kimura et al., 2014**). Another method for homology-directed integration called GeneWeld (**Wierson et al., 2020**) was adapted to include a secondary reporter that, together with the *QF2* sequence, was integrated into exon 1 of the *rln3a* gene to produce *Tg(rln3a:QF2, he1.1:YFP)^c836^*. Identification of *rln3a:QF2* transgenic carriers was facilitated by inclusion of a reporter consisting of a promoter from the *hatching enzyme 1, tandem duplicate 1* (*he1.1*) gene (**Xie et al., 2012**) driving expression of yellow fluorescent protein (YFP) in hatching gland cells starting at 1 day post-fertilization (dpf). Because labeling is transient, the *he1.1:YFP* secondary reporter does not interfere with brain imaging of older larvae.

We confirmed that the *Tg(gsc2:QF2)^c721^* and *Tg(rln3a:QF2, he1.1:YFP)^c836^* driver lines recapitulate endogenous expression patterns of *gsc2* and *rln3a*, respectively, at both larval (**Figure 2B, C, E and F**) and adult (**Figure 1—figure supplement 1A–G**) stages. Consistent with their location in the NI, the *rln3a* and *gsc2* neurons reside on the floor of the 4th ventricle (**Figure 2—figure supplement 1G**), with the *gsc2* neurons anterior to the *rln3a* neurons and also distributed more ventrally up to the dorsal surface of the raphe nucleus (**Figure 2—figure supplement 1C**). Using *TgBAC(gng8:Eco.NfsB-2A-CAAX-GFP)^c375^* to delineate dHb axon terminals at the IPN (**deCarvalho et al., 2013**), we confirmed that *gsc2* neurons are located outside of the IPN in the adult brain, although a few scattered *gsc2* neurons lie just posterior and lateral to it (**Figure 2—figure supplement 2A–D'**). As the

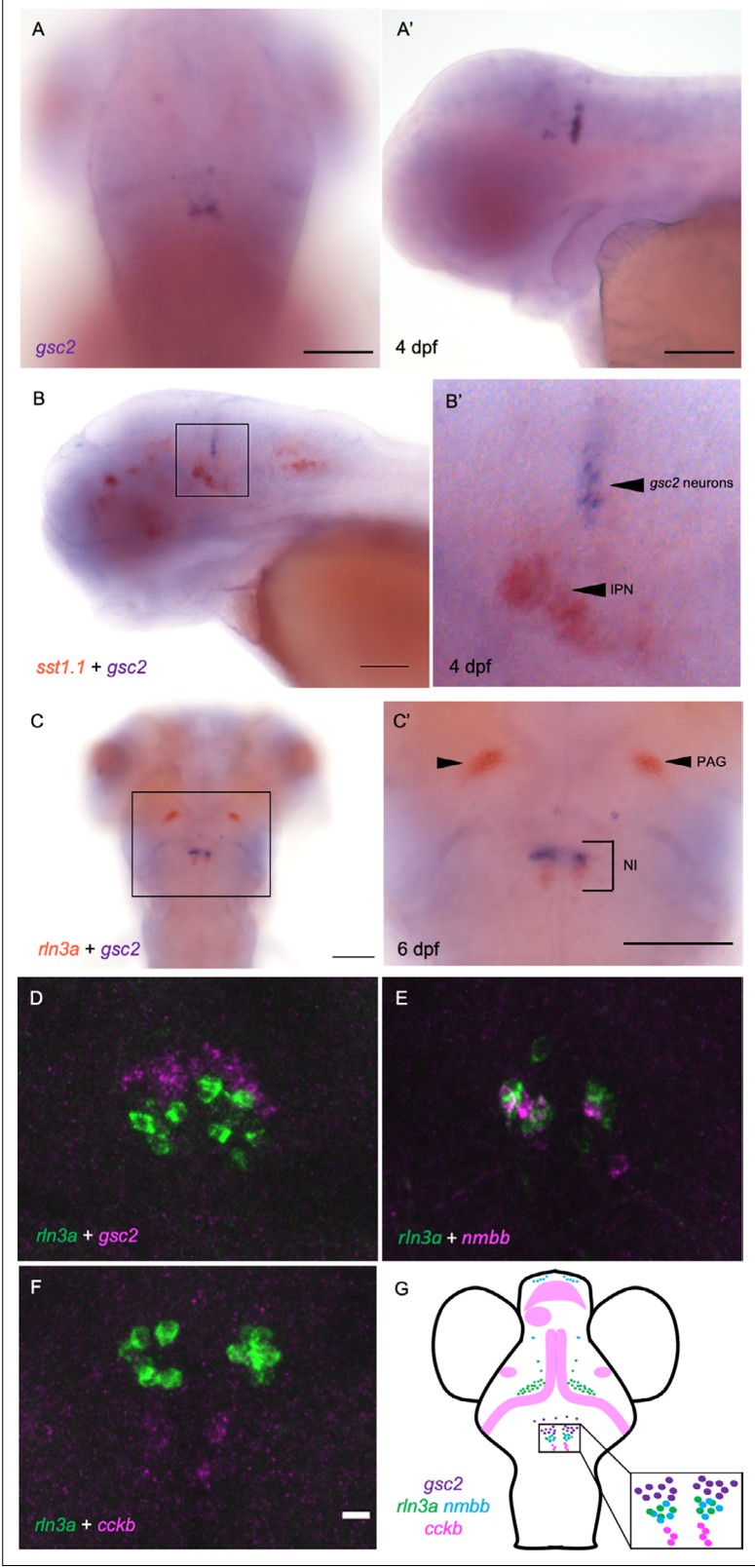

**Figure 1.** *gsc2* neurons localize to the nucleus incertus. (**A, A'**) WISH for *gsc2* and (**B-C'**) double-label WISH for (**B, B'**) *gsc2* and *sst1.1* or (**C, C'**) *gsc2* and *rln3a* was performed on (**A-B'**) 4 days post-fertilization (dpf) or (**C, C'**) 6 dpf larvae. (**A, C, C'**) Dorsal views, anterior to the top. (**A', B, B'**) Lateral views, anterior left. (**B', C'**) Enlarged views of boxed regions in B and C, respectively. Scale bars, 100 μm. (**D–F**) Fluorescent double-label WISH for (**D**) *rln3a* and

*Figure 1 continued on next page*

*Figure 1 continued*

*gsc2*, (**E**) *rln3a* and *nmbb*, and (**F**) *rln3a* and *cckb*. Dorsal views of 6 dpf larvae, anterior to the top. Z-projections. Scale bar, 10 µm. (**G**) Schematic depicting distribution of neuronal subtypes in the nucelus incertus (NI) of larval zebrafish. Green dots, *gsc2* expression; purple dots, *rln3a* expression; blue dots, *nmbb* expression; pink dots and shading, *cckb* expression. IPN: interpeduncular nucleus, PAG: periaqueductal grey, NI: nucleus incertus.

The online version of this article includes the following figure supplement(s) for figure 1:

**Figure supplement 1.** Subset of neuropeptides expressed in NI of larval zebrafish.

**Figure supplement 2.** Partially overlapping expression of *rln3a* and *nmbb* in the zebrafish NI.

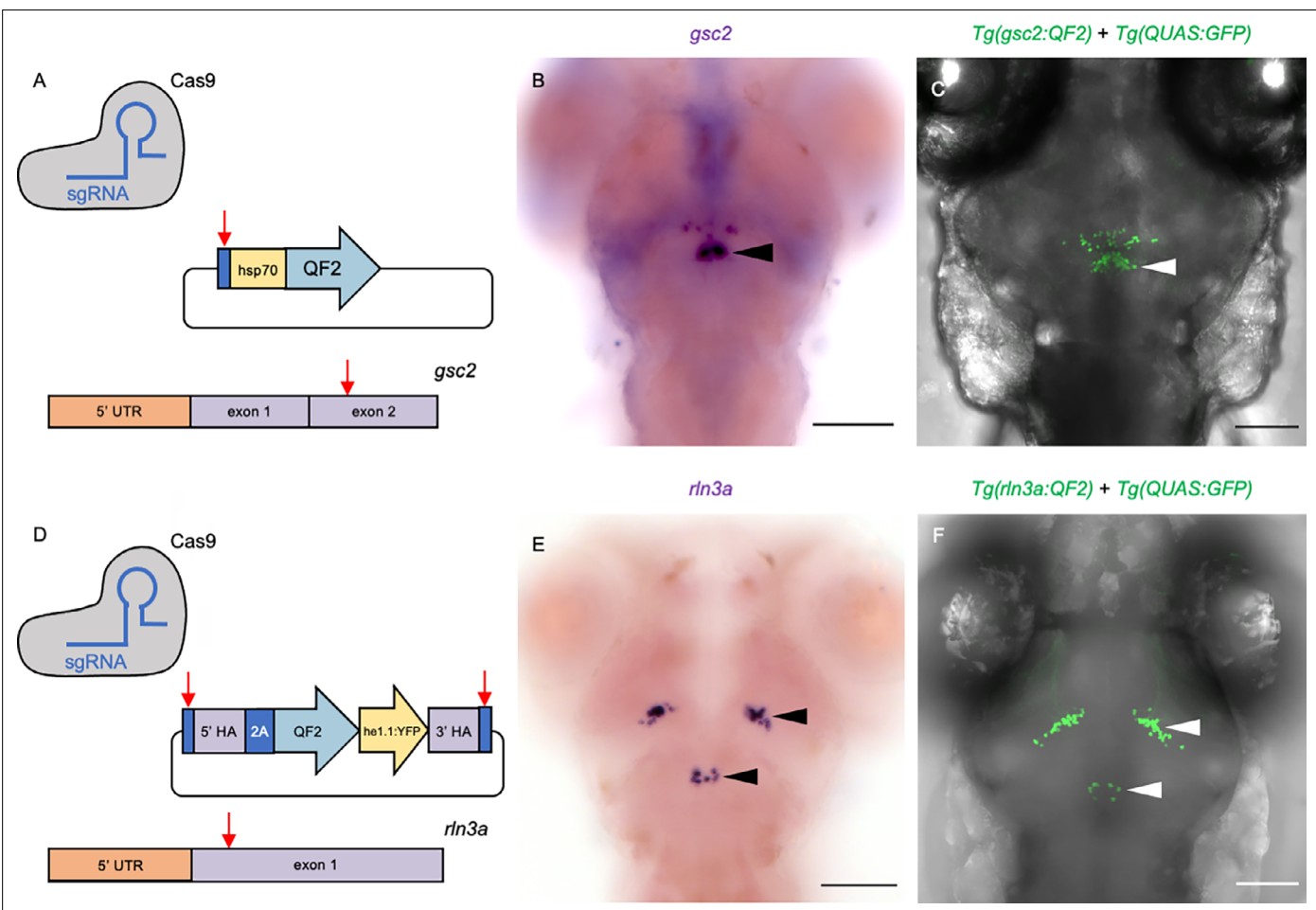

**Figure 2.** Transgenic driver lines recapitulate *gsc2* and *rln3a* expression patterns. (**A, D**) CRISPR/Cas9 genome editing strategies used to generate (**A**) *Tg(gsc2:QF2)$^{c721}$* and (**D**) *Tg(rln3a:QF2, he1.1:YFP)$^{c836}$* driver lines. (**B, C, E, F**) Dorsal views of 6 dpf larvae, anterior to the top. (**B, E**) WISH for (**B**) *gsc2* and (**E**) *rln3a*. (**C, F**) Confocal Z-projections of (**C**) *Tg(gsc2:QF2)$^{c721}$; Tg(QUAS:GFP)$^{c578}$* and (**F**) *Tg(rln3a:QF2, he1.1:YFP)$^{c836}$; Tg(QUAS:GFP)$^{c578}$* larvae. Scale bars, 100 µm. sgRNA: single guide RNA, *hsp70: heat shock cognate 70-kd protein, tandem duplicate 1* promoter, 5' UTR: 5' untranslated region, HA: homology arm, *he1.1*: promoter of hatching enzyme gene.

The online version of this article includes the following figure supplement(s) for figure 2:

**Figure supplement 1.** QF2 driver lines recapitulate *gsc2* and *rln3a* expression patterns in the adult brain.

**Figure supplement 2.** *gsc2* neurons reside outside the IPN in the adult brain.

sparsely distributed anterior group of *gsc2* neurons are anatomically distinct from the main cluster, and not within the nucleus incertus proper (*Figure 2B and C*), they were excluded from subsequent analyses.

## Neurotransmitter identity of *gsc2* and *rln3a* neurons

In mice (*Szőnyi et al., 2019*) and rats (*Olucha-Bordonau et al., 2003*), the NI contains a large population of GABAergic neurons, and *rln3a* NI neurons are largely GABAergic (*Ma et al., 2007*; *Nasirova et al., 2020*). To determine the neurotransmitter identity of the zebrafish *rln3a* and *gsc2* neurons, we mated doubly transgenic fish bearing *Tg(gsc2:QF2)$^{c721}$* or *Tg(rln3a:QF2, he1.1:YFP)$^{c836}$* and a QUAS reporter to fish with transgenes that either label glutamatergic neurons expressing the *solute carrier family 17 member 6b* (*slc17a6b*) gene (*Miyasaka et al., 2009*) or GABAergic neurons expressing *glutamate decarboxylase 1b* (*gad1b*) (*Satou et al., 2013*). We did not observe co-expression of *gsc2* (*Figure 3A*) or *rln3a* (*Figure 3C*) with the glutamatergic reporter in the NI. In contrast, an average of 82.43±3.52% of neurons co-expressed GFP and mApple-CAAX in *TgBac(gad1b:GFP)$^{nns25}$*; *Tg(gsc2:QF2)$^{c721}$*; *Tg(QUAS:mApple-CAAX, he1.1:mCherry)$^{c636}$* larvae (*Figure 3D, D' and G*). Similarly, in *TgBac(gad1b:GFP)$^{nns25}$*; *Tg(rln3a:QF2, he1.1:YFP)$^{c836}$*; *Tg(QUAS:mApple, he1.1:CFP)$^{c788}$* larvae, an average of 80.57±5.57% of neurons co-expressed GFP and mApple (*Figure 3F, F', F" and G*). These results indicate that *gsc2* and *rln3a* neurons are predominantly GABAergic, consistent with their NI identity.

To our knowledge, it has not been verified whether *rln3a* neurons in the periaqueductal gray are also GABAergic. We found that *rln3a* neurons in the PAG were not labeled by the glutamatergic reporter (*Figure 3B*), whereas an average of 81.67±3.81% showed labeling from the *gad1b* transgene (*Figure 3E, E', E" and G*). This suggests that *rln3a* neurons possess similar neurotransmitter identity across neuroanatomical locations.

## Distinct projection patterns of *gsc2* and *rln3a* neurons

To compare the projection patterns of *gsc2* and *rln3a* NI neurons, we expressed membrane-tagged fluorescent reporters in each group and acquired optical sections of their labeled processes using confocal microscopy. At 6 dpf, projections from *gsc2* neurons were prominent in the cerebellum, IPN, raphe, diencephalon, and rostral and caudal hypothalamus (*Figure 4—video 1*, *Figure 4A–E*, *Figure 2—figure supplement 2B, B'*). Sparse efferents from gsc2 neurons were also found in the medulla (*Figure 4—video 1*) and telencephalon (*Figure 4D*). Projections from *rln3a* neurons were found in the medulla, IPN, diencephalon, lateral hypothalamus, and optic tectum (*Figure 4—video 2*, *Figure 4F–J*), with some axons appearing to pass through the posterior commissure (*Figure 4G*). Sparse fibers were also observed in the raphe and telencephalon (*Figure 4H and J*).

Innervation of the IPN by *rln3a* neurons originates solely from the NI cluster, whereas the bulk of projections throughout the brain emanate from *rln3a* neurons in the PAG (*Figure 4—video 2*). This was confirmed by two-photon laser ablation of *rln3a* PAG neurons, which greatly reduced fibers in the medulla, diencephalon, hypothalamus and optic tectum, but spared innervation of the IPN (*Figure 4K–M*). Reduction of *rln3a* PAG neuronal projections enabled visualization of those from the NI, which exclusively target the IPN (*Figure 4—video 3*, *Figure 4L*). Accordingly, ablation of *rln3a* neurons solely in the NI eliminated innervation of the IPN without affecting the rest of the *rln3a* neuron projection pattern, including projections to the medulla, diencephalon, hypothalamus and optic tectum (*Figure 4M*). Efferents from *gsc2* neurons were far more extensive than those of *rln3a* NI neurons, and were observed in regions not innervated by any *rln3a* neurons (e.g. cerebellum and caudal hypothalamus). Thus, the closely apposed *gsc2* and *rln3a* NI neurons exhibit divergent and largely non-overlapping projection patterns.

To examine *gsc2* and *rln3a* efferent innervation of the IPN more precisely, we used *TgBAC(gng8:Eco. NfsB-2A-CAAX-GFP)$^{c375}$* or *TgBAC(gng8:GAL4FF)$^{c426}$*; *Tg(UAS-E1B:NTR-mCherry)$^{c264}$* to delineate the IPN by labeled dHb axon terminals. We confirmed the location of *gsc2* and *rln3a* neuronal cell bodies dorsal to the IPN as visualized by nuclear-tagged reporters (*Figure 5A, A', and C, C'*). Using membrane-tagged reporters, we identified projections from both populations to the IPN (*Figure 5B, D and E–H*) and found that they innervate disparate regions. This is more readily observed in sections of the adult brain in which axons of *gsc2* neurons terminate at the ventral IPN mainly along the midline

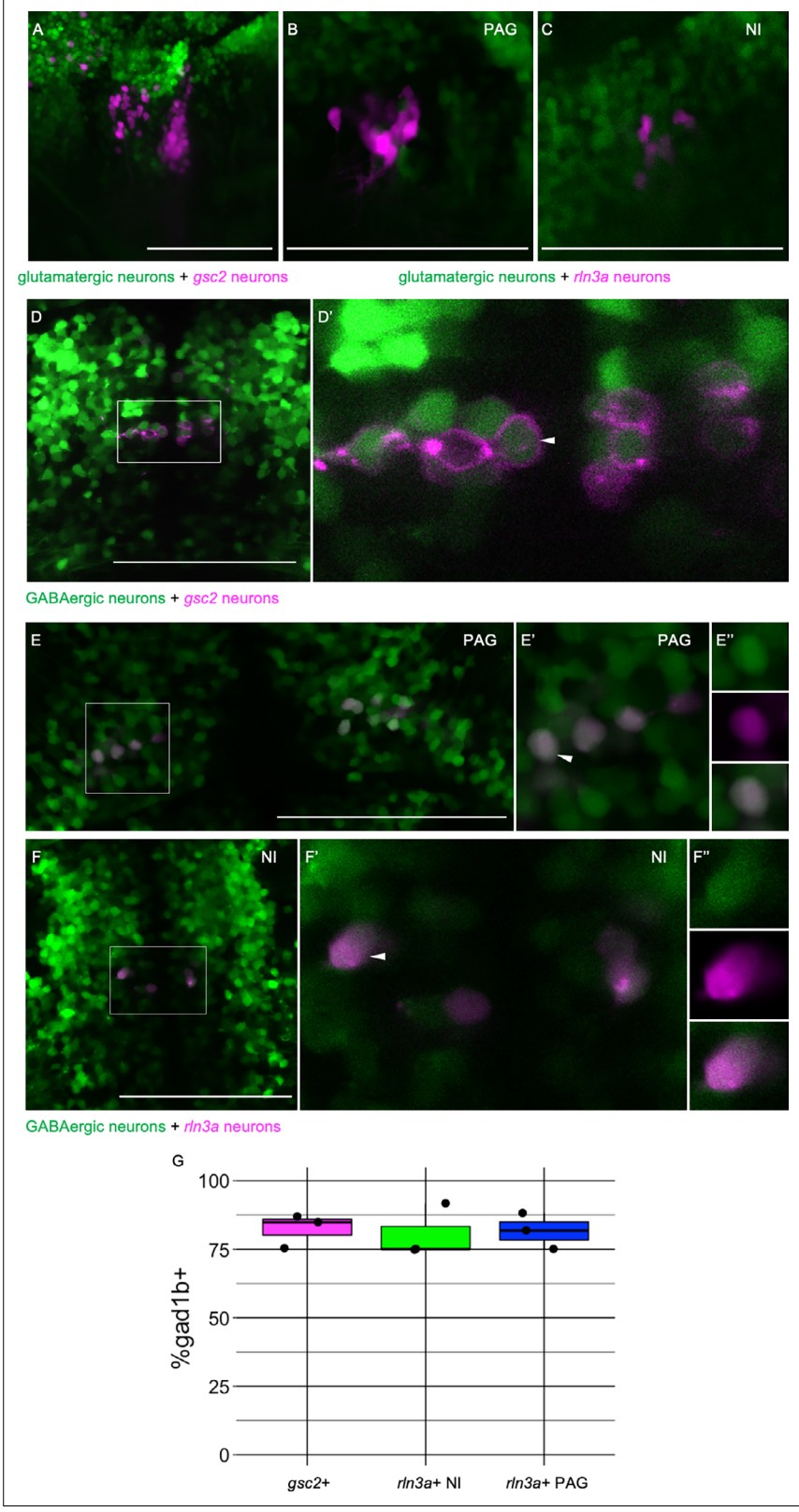

**Figure 3.** *rln3a* and *gsc2* NI neurons are largely GABAergic. (**A-F"**) Confocal images of 6 dpf larvae. (**A–C**) Lateral views, anterior left. (**D-F"**) Dorsal views, anterior to the top. (**A**) Z-projection of *Tg(gsc2:QF2)^{c721}; Tg(QUAS:GFP)^{c578}; Tg(slc17a6b:DsRed)^{nns9}* larva. (**B-F"**) Optical sections. (**B**) PAG and (**C**) NI of a *Tg(rln3a:QF2, he1.1:YFP)^{c836}; Tg(QUAS:mApple, he1.1:CFP)^{c788}; Tg(slc17a6b:GFP)^{zf139}* larva. (**D**) *Tg(gsc2:QF2)^{c721}; Tg(QUAS:mApple-CAAX,*

*Figure 3 continued on next page*

*Figure 3 continued*

*he1.1:mCherry)^{c636}; TgBac(gad1b:GFP)^{nns25} larva.* (**D'**) Magnified view of boxed region in D. White arrowhead indicates a *gad1b* and *gsc2* co-expressing neuron. (**E-F''**) *Tg(rln3a:QF2, he1.1:YFP)^{c836}; Tg(QUAS:mApple, he1.1:CFP)^{c788}; TgBac(gad1b:GFP)^{nns25} larva.* (**E-E''**) View of PAG. (**F-F''**) View of NI. (**E', F'**) Magnified views of boxed regions in E and F. (**E'', F''**) Individual neurons indicated by arrowheads in E' and F', respectively. Top panels: GABAergic, middle panels: *rln3a*, bottom panels: composite. (**G**) Boxplot showing the percentage of *gsc2* and *rln3a* NI neurons, and *rln3a* PAG neurons that express *TgBac(gad1b:GFP)^{nns25}*, n=3 larvae. Scale bars, 100 μm.

neuropil (*Figure 5B, E–F', I*) and axons of *rln3a* neurons terminate at the dorsal IPN (*Figure 5D, G–H', J*), as depicted schematically in *Figure 5K*.

## Afferent input to the NI from the dHb-IPN pathway

Tracing studies in mice (*Lu et al., 2020*), rats (*Goto et al., 2001*; *Olucha-Bordonau et al., 2003*), and zebrafish (*Agetsuma et al., 2010*) suggest that NI neurons receive afferent input from the Hb-IPN pathway. However, it is unclear whether the Hb-IPN axis influences all NI neurons or specific populations.

To test whether the *gsc2* or *rln3a* NI neurons are regulated by the dHb-IPN network, we optogenetically activated the red-shifted opsin ReaChR (*Lin et al., 2013*; *Wee et al., 2019*) in dHb neurons using 561 nm light, while recording calcium transients in either *gsc2* or *rln3a* NI neurons under 488 nm light (*Figure 6A*). We used *Tg(UAS:ReaChR-RFP)^{jf50}* to express ReaChR under control of *TgBAC(gng8:GAL4FF)^{c426}*, which labels dHb neurons that project to the IPN (*Hong et al., 2013*). To verify successful activation of dHb neurons by ReaChR, we also included *Tg(UAS:GCaMP7a)^{zf415}* (*Muto et al., 2013*) to express the calcium indicator GCaMP7 in dHb neurons (*Figure 6B and C*). Simultaneously, we used *Tg(QUAS:GCaMP7a)^{c594}* to express GCaMP7a in either *gsc2* or *rln3a* neurons under control of *Tg(gsc2:QF2)^{c721}* or *Tg(rln3a:QF2, he1.1:YFP)^{c836}* (*Figure 6B' and C'*).

We first confirmed that 561 nm light increases calcium signaling in ReaChR-expressing dHb neurons, but not in ReaChR-negative controls (*Figure 6D–D'' and F–F''*, *Figure 6—figure supplement 1A, D, G, J*). Next, we showed that ReaChR activation in the dHb increased calcium transients in *gsc2* NI neurons, as they showed greater activation in response to 561 nm light in ReaChR-expressing larvae than in ReaChR-negative controls (*Figure 6E–E''*, *Figure 6—figure supplement 1B, C, E, F*). However, similar levels of calcium signaling were detected in the *rln3a* NI neurons of ReaChR-expressing larvae and negative controls in response to 561 nm light (*Figure 6G–G''*, *Figure 6—figure supplement 1H, I, K, L,*). Statistically significant differences in the activation of *rln3a* PAG neurons between ReaChR-expressing larvae and ReaChR-negative controls were also not detected (*Figure 6H–H''*, *Figure 6—figure supplement 1M, N, O, P*). These results show that activation of the dHb-IPN axis increases activity in *gsc2* neurons but not in *rln3a* neurons, indicating that the latter do not directly mediate functions of the dHb-IPN pathway.

## Spontaneous and evoked activity differs between *gsc2* and *rln3a* neurons

In rodents, aversive stimuli, such as foot shock, air puff, water-restraint stress, exposure to an elevated plus maze, and the anxiogenic drug FG-7142 all increase neuronal activity in the NI (*Lawther et al., 2015*; *Lu et al., 2020*; *Passerin et al., 2000*; *Rajkumar et al., 2016*; *Szőnyi et al., 2019*; *Tanaka et al., 2005*), yet whether NI neuronal subtypes show distinct responses to aversive stimuli is unclear. To determine whether *gsc2* and *rln3a* neurons differ in their response to an aversive stimulus, we recorded calcium transients upon delivery of a mild electric shock (25 V, 200 ms duration) (*Duboué et al., 2017*) to immobilized larvae (*Figure 7A, B and C*). The *gsc2* neurons showed little spontaneous activity and a robust increase in calcium signaling in response to shock (*Figure 7D and D'*, *Figure 7— video 1*). By contrast, *rln3a* neurons showed more spontaneous fluctuations in activity throughout the recording period (*Figure 7E–H*, *Figure 7—video 1*), producing a wider distribution of amplitudes (*Figure 7F–G'*), and their response to shock was shorter in duration than for *gsc2* neurons (*Figure 7I*). The *gsc2* and *rln3a* neurons therefore differ in their spontaneous activity and in the duration of their response to an aversive stimulus.

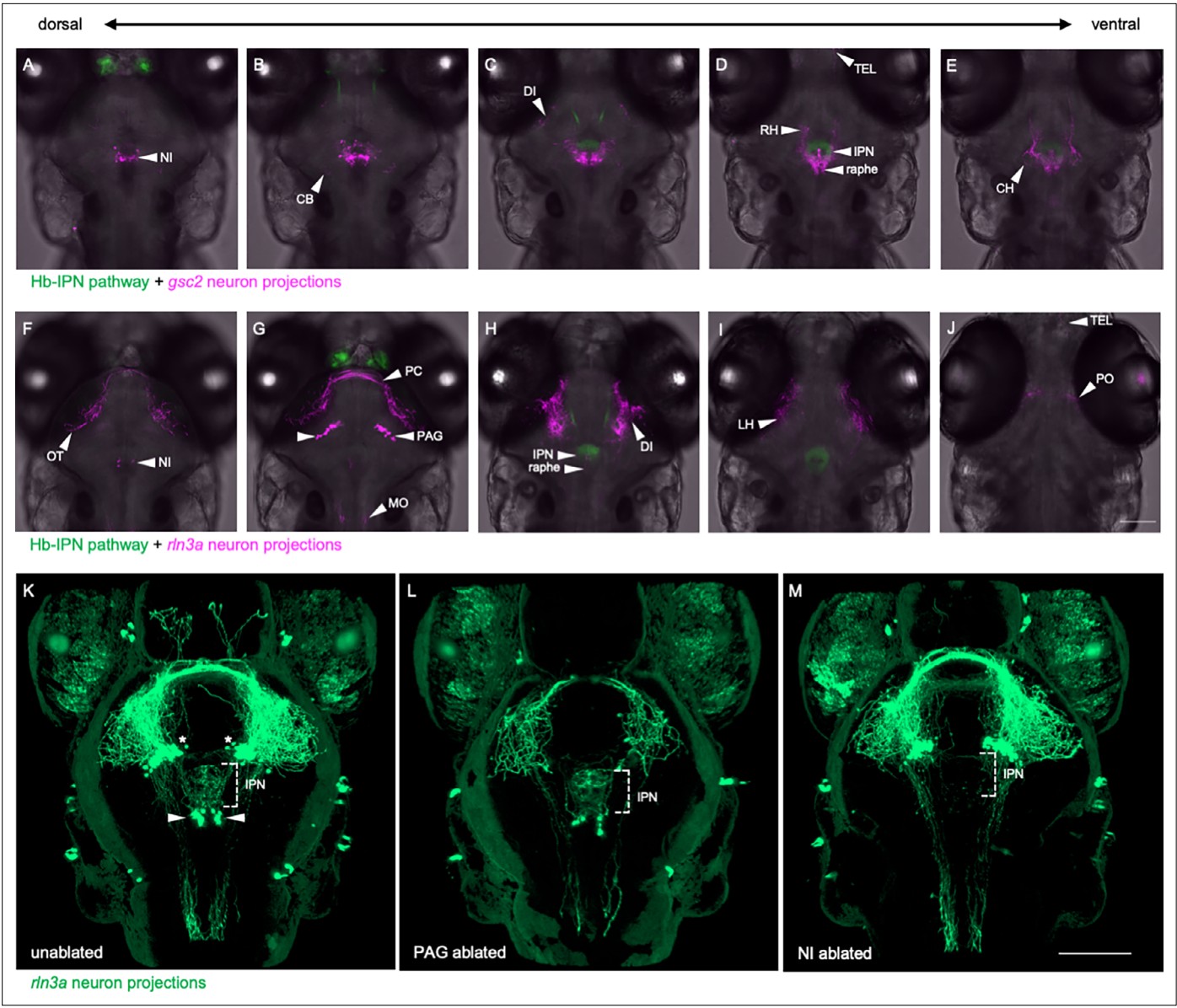

**Figure 4.** Distinct projection patterns of *gsc2* and *rln3a* neurons. (**A–J**) Confocal optical sections of (**A–E**) *Tg(gng8:Eco.NfsB-2A-CAAX-GFP)*^c375^; *Tg(gsc2:QF2)*^c721^; *Tg(QUAS:mApple-CAAX, he1.1:mCherry)*^c636^ and (**F–J**) *Tg(gng8:Eco.NfsB-2A-CAAX-GFP)*^c375^; *Tg(rln3a:QF2, he1.1:YFP)*^c836^; *Tg(QUAS:mApple-CAAX, he1.1:mCherry)*^c636^ 6 dpf larvae ordered from dorsal to ventral. (**K–M**) 3D reconstructions of confocal Z-stacks generated using Zen software (Zeiss), *Tg(rln3a:QF2, he1.1:YFP)*^c836^; *Tg(QUAS:GFP-CAAX)*^c591^; *Tg(QUAS:NLS-GFP, he1.1:CFP)*^c682^ larvae at 7 dpf showing efferents from (**K**) intact *rln3a* PAG (asterisks) and NI (arrows) neurons or following two-photon laser-mediated ablation of (**L**) PAG or (**M**) NI *rln3a* cell bodies at 6 dpf. Dorsal views, anterior to the top. Scale bars, 100 µm. NI: nucleus incertus, OT: optic tectum, CB: cerebellum, PC: posterior commissure, PAG: periaqueductal gray, MO: medulla oblongata, DI: diencephalon, IPN: interpeduncular nucleus, TEL: telencephalon, RH: rostral hypothalamus, LH: lateral hypothalamus, CH: caudal hypothalamus, PO: pre-optic area.

The online version of this article includes the following video(s) for figure 4:

**Figure 4—video 1.** Axonal projections of *gsc2* neurons.

https://elifesciences.org/articles/89516/figures#fig4video1

**Figure 4—video 2.** Axonal projections of *rln3a* neurons.

https://elifesciences.org/articles/89516/figures#fig4video2

**Figure 4—video 3.** Axonal projections of *rln3a* neurons after ablation of *rln3a* PAG cell bodies.

https://elifesciences.org/articles/89516/figures#fig4video3

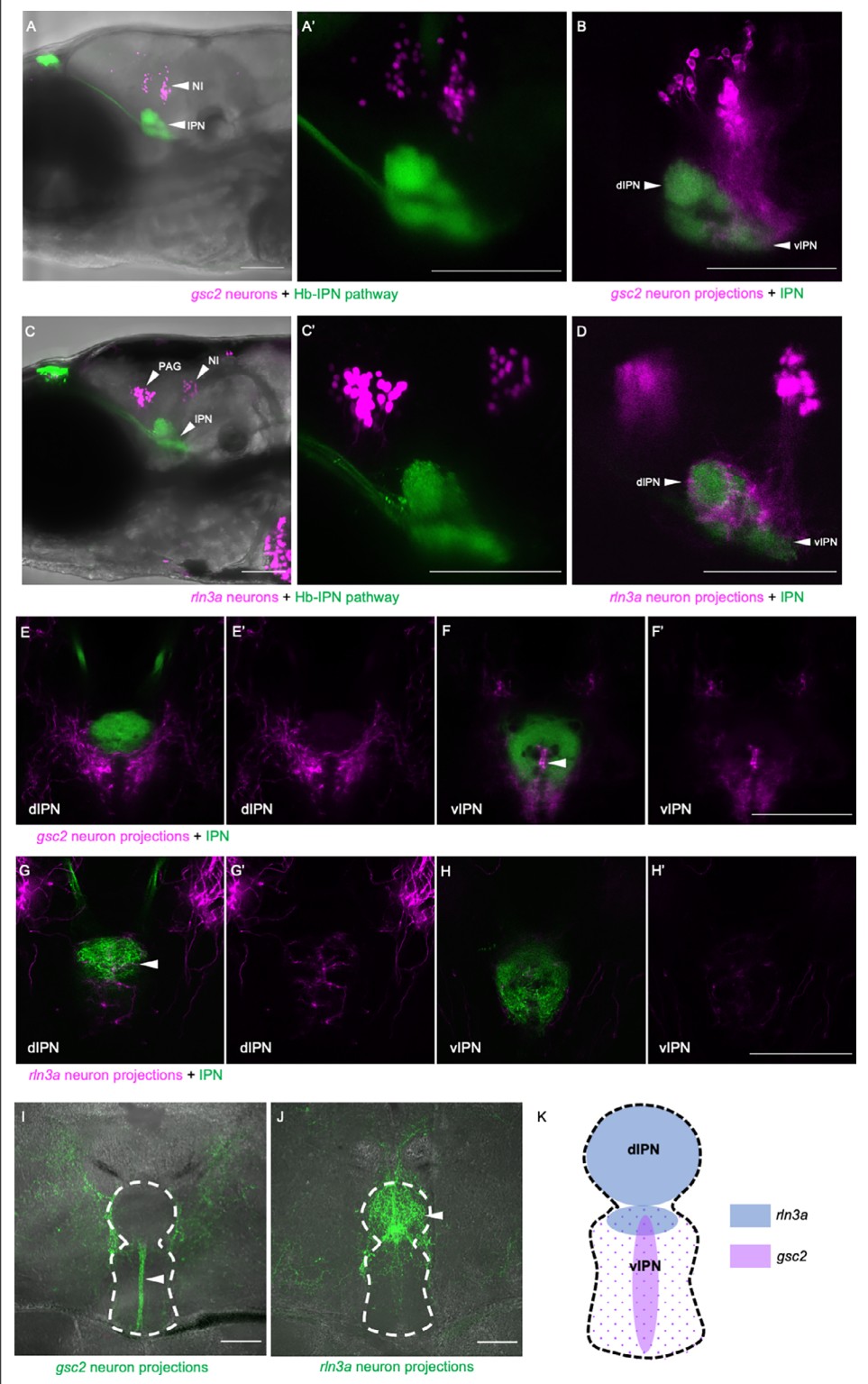

**Figure 5.** *gsc2* and *rln3a* NI neurons innervate different dorsoventral IPN regions. (**A-H'**) Confocal images of 6 dpf larvae. (**A-B, E-F'**) *TgBAC(gng8:Eco.NfsB-2A-CAAX-GFP)^c375* and *Tg(gsc2:QF2)^c721* driving (**A, A'**) *Tg(QUAS:NLS-mApple, he1.1:CFP)^c718* or (**B, E-F'**) *Tg(QUAS:mApple-CAAX, he1.1:mCherry)^c636*. (**C-D, G-H'**) *TgBAC(gng8:GAL4FF)^c426; Tg(UAS-E1B:NTR-mCherry)^c264* and *Tg(rln3a:QF2, he1.1:YFP)^c836* driving (**C, C'**) *Tg(QUAS:NLS-GFP, he1.1:CFP)^c682* or (**D, G-H'**) *Tg(QUAS:NLS-GFP, he1.1:CFP)^c682* and *Tg(QUAS:GFP-CAAX)^c591*. (**A', C'**) Higher magnification images of larvae in A and C, respectively. (**A, A', C, C'**) Z-projections. (**B, D**) optical

*Figure 5 continued on next page*

*Figure 5 continued*

sections. (**A–D**) Lateral views, anterior left. (**E-H'**) Dorsal views, anterior to the top. Optical sections at the level of the (**E, E', G, G'**) dorsal IPN or (**F, F', H, H'**) ventral IPN of the same larvae. (**E', F', G', H'**) Labeled efferent projections only. (**I, J**) Confocal Z-projections of coronal sections (70 µm) through adult brains of (**I**) *Tg(gsc2:QF2)*[c721]; *Tg(QUAS:GFP-CAAX; he1.1:YFP)*[c631] or (**J**) *Tg(rln3a:QF2; he1.1:YFP)*[c836]; *Tg(QUAS:GFP-CAAX)*[c591] fish. Anterior to the top. (**K**) Schematic of the IPN showing distinct dorsoventral regions innervated by *rln3a* and *gsc2* neurons. Scale bars, 100 µm. dIPN: dorsal IPN, vIPN: ventral IPN.

## Ablation of *rln3a* but not *gsc2* neurons alters locomotor activity

Previous reports have implicated the NI in regulating locomotor activity and proposed that an animal's increased movement after an aversive stimulus is, in part, mediated by increased activity in the NI (*Farooq et al., 2016*; *Lu et al., 2020*). We tested whether eliminating small populations of NI neurons (i.e. 10.67±1.33 *rln3a* neurons or 48.33±2.33 *gsc2* neurons) would be sufficient to influence baseline locomotor behavior or the response to electric shock, which normally elicits immediate hyperactivity in larval zebrafish (*Duboué et al., 2017*).

With GFP expression as a guide, we used a two-photon laser to selectively ablate *gsc2* (*Figure 8A and A'*), *rln3a* neurons in the NI (*Figure 8B and B'*), or *rln3a* neurons in the PAG (*Figure 8C–C'''*), at 6 dpf. We confirmed ablation by WISH (*Figure 8—figure supplement 1A,, A', D, D'*), and also verified selectivity by determining that *rln3a* NI neurons were spared in larvae with ablated *gsc2* neurons (*Figure 8—figure supplement 1B, B'*), and, conversely, that *gsc2* neurons were intact in larvae with ablated *rln3a* NI neurons (*Figure 8—figure supplement 1C, C'*). One day later (7 dpf), we tracked locomotion of individual freely swimming ablated larvae and unablated siblings for 2 min. After recording baseline activity, we delivered a single electric shock (25 V, 200 ms duration) to each larva and measured the locomotor response (*Duboué et al., 2017*).

Both ablated and unablated larvae exhibited hyperactivity immediately following shock (*Figure 8D–E*), and statistically significant differences in the response to shock were not detected (*Figure 8E*). This suggests that neither the *gsc2* neurons nor the neighboring *rln3a* neurons are required for the immediate behavioral response to shock. Unexpectedly, however, larvae that lacked *rln3a* NI neurons swam a greater overall distance during the pre-shock period and exhibited longer phases of activity than unablated controls, larvae with ablated *gsc2* neurons, or larvae with ablated *rln3a* PAG neurons (*Figure 8F–H*, *Figure 8—video 1*). Phases of activity were defined by continuous movement of the larvae with no more than one second of prolonged immobility. The total number of phases was similar in all groups (*Figure 8I*). This indicates that ablation of *rln3a* NI neurons promotes prolonged phases of movement, rather than increasing the frequency of movement initiation. In addition to increased swimming, larvae that lacked *rln3a* NI neurons showed increased turning behavior. For larvae with ablated *rln3a* NI neurons, the change in angle of orientation measured per unit of distance traveled was greater than that of unablated controls, larvae with ablated *gsc2* neurons, or larvae with ablated *rln3a* PAG neurons (*Figure 8-figure supplement 2A*). The unablated control larvae included both *Tg(gsc2:QF2)*[c721]; *Tg(QUAS:GFP)*[c578] and *Tg(rln3a:QF2, he1.1:YFP)*[c836]; *Tg(QUAS:GFP)*[c578] siblings of ablated larvae, although repeating the analyses with the control group containing only one genotype or the other did not change the conclusions (*Figure 8-figure supplement 2B, C* and *Figure 8-figure supplement 3*). Overall, the results show that spontaneous locomotor activity and turning behavior are increased following ablation of the *rln3a* cluster of NI neurons, whereas swimming behavior is normal after ablation of the larger *gsc2* population.

## Discussion

The nucleus incertus ('uncertain nucleus'), first described in the human brain in 1903 (*Streeter, 1903*), remains an enigmatic structure that has been implicated in stress (*Bittencourt and Sawchenko, 2000*; *Lawther et al., 2015*; *Passerin et al., 2000*; *Potter et al., 1994*; *Rajkumar et al., 2016*; *Tanaka et al., 2005*), arousal (*Lu et al., 2020*), and memory (*Ma et al., 2009*; *Szőnyi et al., 2019*). As the NI is the primary source of relaxin-3 expressing neurons in the rodent brain, they have been a focus of interest even though not all NI neurons produce this neuropeptide (*Ma et al., 2013*; *Nasirova et al., 2020*). Here, we compare the properties of the cells expressing *rln3a* with a neighboring group of neurons in the zebrafish NI.

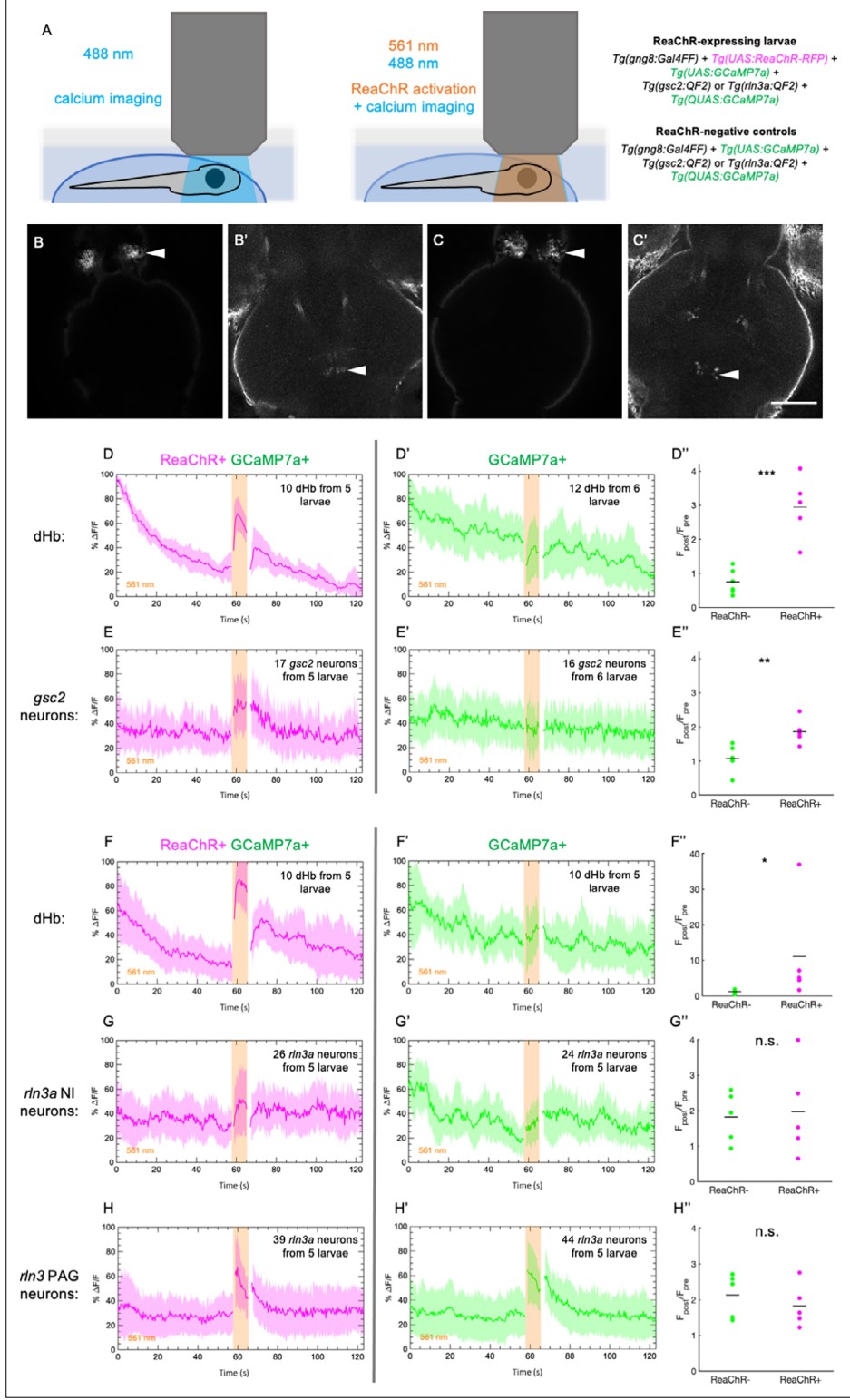

**Figure 6.** Increased calcium signaling in *gsc2* neurons upon optogenetic activation of the dHb. Calcium transients were imaged at 2.6 Hz before, during, and after illumination with 561 nm light in 7 dpf larvae. (A) Drawings depicting imaging of calcium transients and optogenetic activation using confocal microscopy. (B-C') Representative maximum intensity projections of *GCaMP7a* fluorescence in (B) dHb and (B') *gsc2* neurons of the same larva, or (C) dHb and (C') *rln3a* NI neurons of the same larva. Anterior to the top. Scale bar, 100 μm. (D-E") *Tg(gsc2:QF2)^c721* or (F-H") *Tg(rln3a:QF2, he1.1:YFP)^c836* driver lines in (D-H) *TgBAC(gng8:GAL4FF)^c426*;

*Figure 6 continued on next page*

*Figure 6 continued*

*Tg(UAS:GCaMP7a)$^{zf415}$; Tg(QUAS:GCaMP7a)$^{c594}$* larvae (D, E, F, G, H) with or (D', E', F', G', H') without *Tg(UAS:ReaChR-RFP)$^{jf50}$*. The average change in GCaMP7a signaling (%ΔF/F) is shown for (D, D', F, F') the dorsal habenulae, (E, E') *gsc2* neurons, (G, G') *rln3a* NI neurons, and (H, H') *rln3a* PAG neurons. Shading indicates standard deviation. Gaps at light onset and offset are due to latency in switching the laser configuration. (D", E", F", G", H") Average $F_{post}/F_{pre}$ is shown for (D", F") the dHb, (E") *gsc2* neurons, (G") *rln3a* NI neurons, and (H") *rln3a* PAG neurons of *ReaChR$^+$* and *ReaChR$^-$* larvae. $F_{post}$ is the area under the curve for 15 frames (5.8 s) during 561 nm illumination and $F_{pre}$ is the area under the curve for 15 frames (5.8 s) preceding 561 nm illumination. (D", E", F", G", H") Black bars indicate mean ratios: (D") 0.75±0.15, n=6 *ReaChR$^-$* larvae, 2.95±0.41, n=5 *ReaChR$^+$* larvae, ***p=0.0004. (E") 1.07±0.15, n=6 *ReaChR$^-$* larvae, 1.86±0.17, n=5 *ReaChR$^+$* larvae, **P=0.0073. (F") 1.22±0.29, n=5 *ReaChR$^-$* larvae, 11.08±6.54, n=5 *ReaChR$^+$* larvae, *p=0.032. (G") 1.82±0.32, n=5 *ReaChR$^-$* larvae, 1.97±0.59, n=5 *ReaChR$^+$* larvae, = 0.83. (H") 2.13±0.27, n=5 *ReaChR$^-$* larvae, 1.83±0.27, n=5 *ReaChR$^+$* larvae, p=0.45. Extended y-axis in F" to display higher values.

The online version of this article includes the following figure supplement(s) for figure 6:

**Figure supplement 1.** Calcium signaling in individual larvae and neurons.

Through transcriptional profiling, *gsc2* transcripts were found to be enriched in samples dissected from the adult zebrafish brain that encompassed the IPN. However, closer examination of both larval and adult brains revealed that *gsc2*-expressing neurons are located outside of the IPN, just anterior to the *rln3a* neurons in the NI. The enriched transcripts were likely due to the presence of *gsc2*-positive neurons that lie just posterior and lateral to the IPN. Neurons expressing the *Gsc2* murine homolog had previously been identified in the mouse brain, although there is conflicting information about their precise anatomical location (*Funato et al., 2010*; *Gong et al., 2003*; *Gottlieb et al., 1998 Saint-Jore et al., 1998*). On the basis of our results, we suspect that *Gsc2* neurons are not located within the rodent IPN as was previously concluded (*Funato et al., 2010*; *Gong et al., 2003*), but are likely situated adjacent to it.

We developed transgenic tools to characterize *gsc2* and *rln3a* neurons in more detail. The Gal4-UAS system of yeast is widely used in zebrafish to express reporter genes in specific cell populations; however, its utility for small groups of neurons is limited because of mosaicism due to progressive methylation of CpG residues in multicopy upstream activation sequences (UAS), resulting in transcriptional silencing (*Goll et al., 2009*). The QF2/QUAS system of *Neurospora* (*Riabinina and Potter, 2016*; *Subedi et al., 2014*), coupled with CRISPR/Cas9 integration, enabled the generation of targeted driver lines and robust and selective expression of reporter genes in either *gsc2* or *rln3a* neurons. By labeling with QUAS-driven fluorescent reporters, we determined that the anatomical location, neurotransmitter phenotype, and hodological properties of *gsc2* and *rln3a* neurons are consistent with NI identity, supporting the assertion that the griseum centrale of fish is analogous to the mammalian NI. Both groups of neurons are GABAergic, reside on the floor of the fourth ventricle and project to the interpeduncular nucleus. However, these adjacent neuronal populations have distinct connections with the IPN and other brain regions, and also differ in their afferent input, calcium signaling, and influence on locomotor behavior (summarized in *Table 1*). Owing that the NI has been proposed to act in concert with the median raphe and IPN, in 'a midline behavior control network of the brainstem' (*Goto et al., 2001*), it is important to build the framework of neuronal subtypes that mediate such coordinated activity.

## The IPN as an integrating center for dHb and NI input

Previous work demonstrated that axons from left dHb and right dHb neurons innervate different regions along the dorsoventral extent of the IPN; neurons in the left dHb project to both the dorsal IPN (dIPN) and ventral IPN (vIPN), whereas right dHb neurons mainly target the vIPN (*Gamse et al., 2005*). We found that different populations of NI neurons also target specific IPN compartments; *rln3a* neurons project mainly to the dIPN and *gsc2* neurons predominantly innervate the vIPN along its midline neuropil. A study by *Zaupa et al., 2021* demonstrates that axon terminals from cholinergic and noncholinergic dHb neurons, which innervate the vIPN and dIPN respectively, show distinct patterns of activity. Spontaneous calcium spikes in cholinergic dHb terminals at the vIPN coincide with transient decreases in calcium signaling in non-cholinergic dHb terminals at the dIPN. This negatively correlated activity is mediated by activation of vIPN neurons that release GABA to inhibit non-cholinergic dHb

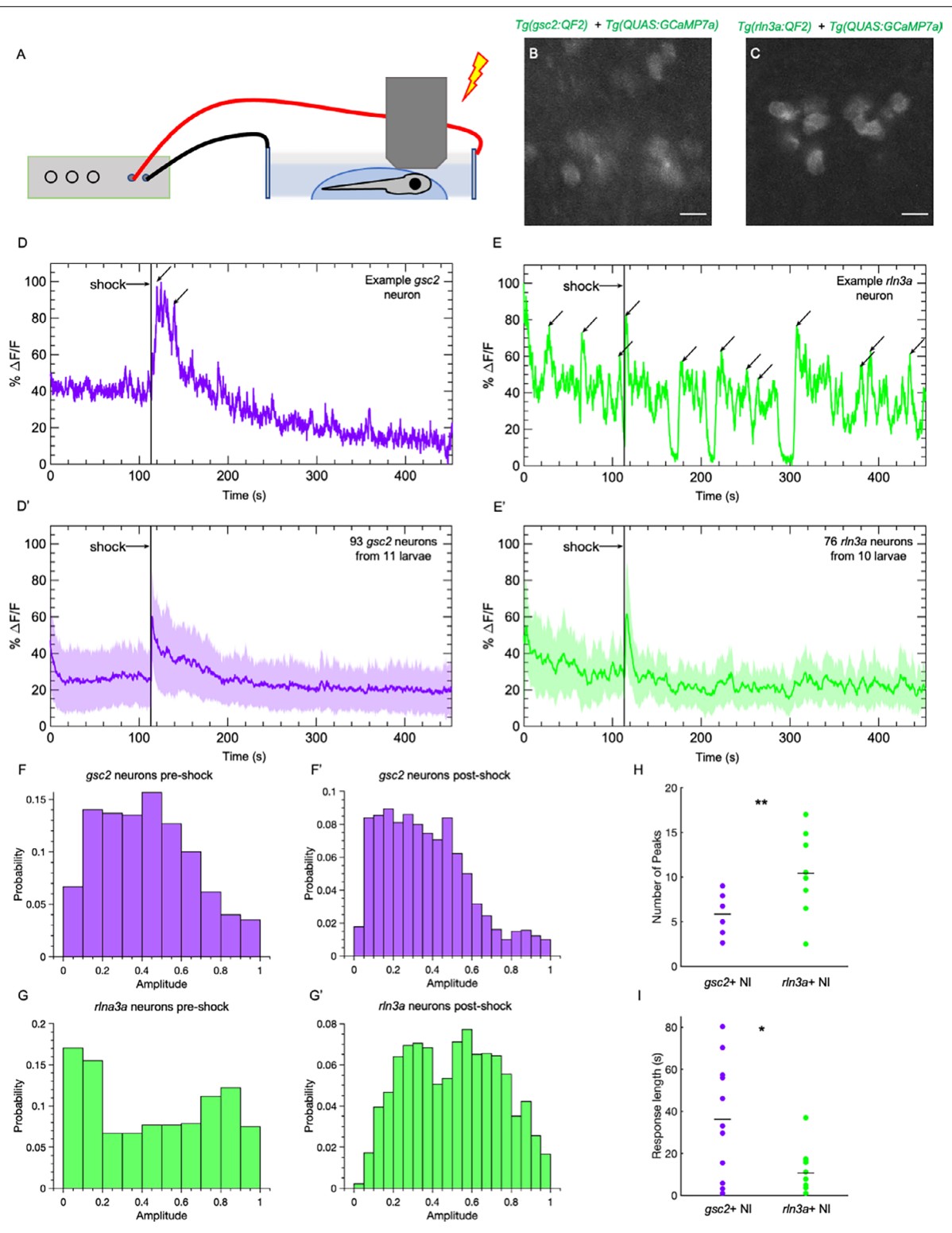

**Figure 7.** *gsc2* and *rln3a* NI neurons differ in their spontaneous activity and response to an aversive cue. Calcium transients were imaged at 5.2 Hz in 7dpf larvae during a mild electric shock (25 V, 200ms duration). (**A**) Drawing depicting delivery of shock to an immobilized larva during imaging. (**B, C**) Examples of maximum intensity projections for NI neurons in (**B**) *Tg(gsc2:QF2)^{c-721}; Tg(QUAS:GCaMP7a)^{c594}* or (**C**) *Tg(rln3a:QF2, he1.1:YFP)^{c836}; Tg(QUAS:GCaMP7a)^{c594}* larvae. Dorsal views, anterior to the top. Scale bars, 10 µm. (**D, E**) GCaMP7a signaling (%ΔF/F) for representative individual (**D**) *gsc2* or (**E**) *rln3a* neurons. Arrows indicate local maxima identified as peaks by the MATLAB *findpeaks* function (MinPeakProminence: 0.3, MinPeakWidth: 10). (**D', E'**) Average %ΔF/F for all recorded (**D'**) *gsc2* neurons (93 from 11 larvae) or (**E'**) *rln3a* neurons (76 from 10 larvae). Shading

*Figure 7 continued on next page*

*Figure 7 continued*

indicates standard deviation. (**F, F′, G, G′**) Histogram of %ΔF/F amplitudes for (**F, F′**) *gsc2* or (**G, G′**) *rln3a* neurons during the (**F, G**) pre-shock or (**F′, G′**) post-shock period. (**H, I**) Average (**H**) number of peaks during the recording period (as depicted by arrows in examples D and E) and average (**I**) length of response for *gsc2* neurons and *rln3a* neurons, defined as the time required for the %ΔF/F to return to a value equal to or less than the average %ΔF/F in the 100 frames (18.9 seconds) prior to shock. Black bars in (**H**) indicate mean peaks for *gsc2* neurons (5.56±0.63, n=11 larvae) and *rln3a* neurons (9.91±1.18, n=10 larvae), **p=0.0035. Black bars in (**I**) indicate mean response times for *gsc2* neurons (36.21±8.42, n=11 larvae) and *rln3a* neurons (10.63±3.27, n=10 larvae) *p=0.045.

The online version of this article includes the following video(s) for figure 7:

**Figure 7—video 1.** Response of *gsc2* neurons to shock.
https://elifesciences.org/articles/89516/figures#fig7video1

**Figure 7—video 2.** Response of *rln3a* neurons to shock.
https://elifesciences.org/articles/89516/figures#fig7video2

terminals at the dIPN through their presynaptic GABA$_B$ receptors. Our results raise the possibility that innervation by different populations of NI neurons also shapes activity in the dorsal and ventral IPN. The IPN could thus integrate signals from disparate neuronal populations in the dHb and NI, and perhaps other brain regions. Future work will explore how the activity of *rln3a* and *gsc2* axon terminals is coordinated with cholinergic and non-cholinergic dHb input to the dorsal and ventral IPN.

## Distinct patterns of calcium signaling by NI neurons

Distinct patterns of activity were observed in the neuronal populations of the NI, with *gsc2* neurons having little spontaneous activity and *rln3a* neurons exhibiting continuous fluctuations in calcium signaling. A study in rats found that relaxin-3 neurons fire in synchrony with the ascending phase of the hippocampal theta oscillation (4–12 Hz), which has been implicated in spatial memory (*Ma et al., 2013*). Stimulation of NI neurons in rats and *Nmb* NI neurons in mice also increases hippocampal theta power (*Lu et al., 2020*; *Nuñez et al., 2006*). The oscillating calcium transients that we detected in *rln3a* neurons of larval zebrafish are on the order of seconds; however, consistent with infra-slow waves that occur at frequencies in the range of tens to hundreds of seconds, and within which fast oscillations are often nested (*Palva and Palva, 2012*). Infra-slow oscillations correlate with rhythmic fluctuations in performance observed in psychophysical experiments with humans, in which a subject performs a task of constant difficulty for several minutes. It has been proposed, therefore, that intra-slow waves coordinate shifts between attentive and inattentive brain states (*Palva and Palva, 2012*). Given that ablation of *rln3a* NI neurons increases the length of phases of movement in zebrafish larvae, fluctuating activity in *rln3a* neurons may control transitions between phases of behavioral activity and inactivity.

## Cell-type-specific roles for the NI

Rodent studies have described the behavior of animals with null mutations in the gene encoding RLN3 (*Smith et al., 2012*), or its receptor, RXFP (*Hosken et al., 2015*), and found decreased voluntary wheel running, suggesting that the relaxin-3 system is involved in regulating locomotor activity. However, it is difficult to attribute mutant phenotypes to specific sub-groups of *Rln3* neurons. Activation of the NI through microstimulation or chemogenetics increased movement in rats (*Farooq et al., 2016*; *Ma et al., 2017*), which implicates the NI region in regulating locomotor activity but does not identify the relevant neurons.

Strikingly, removal of *rln3a* NI neurons elicited hyperactivity in zebrafish larvae. Ablation of *rln3a* neurons in the PAG did not affect locomotion. This suggests that the role of the NI in regulating baseline locomotor activity is mediated by *rln3a* neurons. Because some *nmbb* neurons are interspersed with *rln3a* neurons in the NI, we cannot eliminate the possibility that loss of *nmbb* neurons also contributes to the hyperactivity phenotype. Previous studies in adult rodents indicate that enhanced NI activity promotes locomotion, but we find the opposite in larval zebrafish; NI neurons normally suppress spontaneous locomotor activity. Interestingly, a study of dopaminergic signaling in larval zebrafish also reported that dopamine suppressed spontaneous fictive swim episodes (*Thirumalai and Cline, 2008*), although dopamine is classically known for stimulating locomotor activity in adult rodents (*Ryczko and Dubuc, 2017*). Thus, differential roles for neuromodulators during development and adulthood could be a general feature of locomotor circuitry.

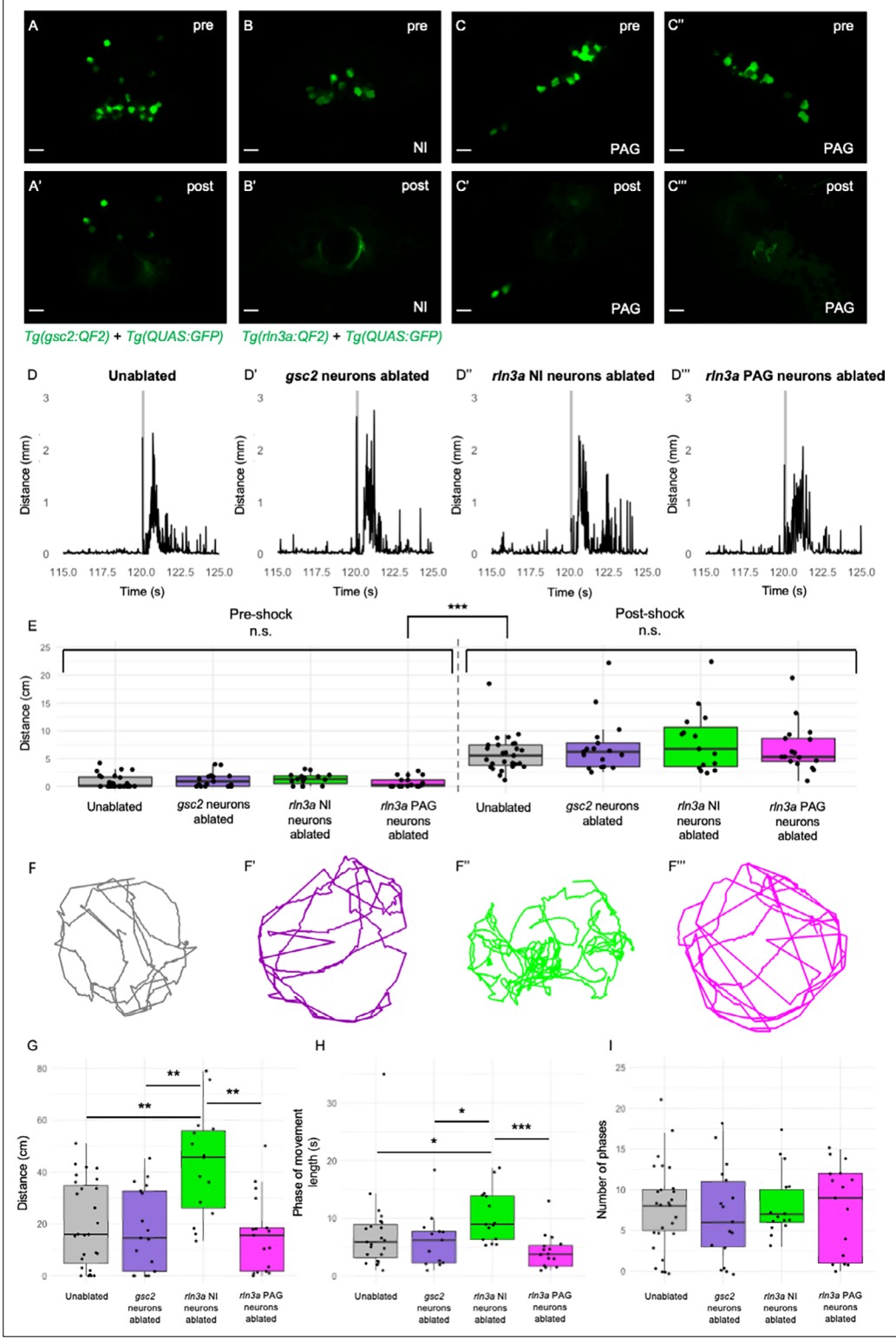

**Figure 8.** Loss of *rln3a* NI neurons increases spontaneous locomotor activity. (**A-C′′′**) Single optical sections from two-photon imaging of 6 dpf (**A, A′**) *Tg(gsc2:QF2)$^{c721}$; Tg(QUAS:GFP)$^{c578}$* or (**B-C′′′**) *Tg(rln3a:QF2, he1.1:YFP)$^{c836}$; Tg(QUAS:GFP)$^{c578}$* larvae (**A, B, C, C′′**) before and (**A′, B′, C′, C′′′**) after laser-mediated ablation of (**A, A′**) *gsc2* neurons, (**B, B′**) *rln3a* NI neurons, or (**C, C′**) left and (**C′′, C′′′**) right *rln3a* PAG neurons. Dorsal views, anterior to the top. Scale bars, 10 μm. (**D-D′′′**) Average locomotor activity during 5 s prior to and after shock. Shock delivery is denoted by the gray line. (**E**) Mean of total distance traveled during 5 s pre- and

*Figure 8 continued on next page*

*Figure 8 continued*

post-shock for unablated controls (pre=0.86±0.23 cm, post=5.89±0.64 cm, n=27), or larvae with ablated *gsc2* (pre=1.19±0.31, post=7.23±1.20 cm, n=17), *rln3a* NI (pre=1.30±0.26 cm, post=8.09±1.45 cm, n=15), or *rln3a* PAG (pre=0.72±0.23 cm, post=6.92±1.07 cm, n=17) neurons. Kruskal-Wallis rank sum test: ***p=2.2 x 10⁻¹⁶. Dunn's post-hoc tests with adjustment for multiple comparisons show no statistically significant differences within pre- and post-shock epochs, p<0.001*** for each pre-shock vs. post-shock comparison. Unablated control group includes *Tg(gsc2:QF2)^c721*; *Tg(QUAS:GFP)^c578* and *Tg(rln3a:QF2, he1.1:YFP)^c836*; *Tg(QUAS:GFP)^c578* siblings of ablated larvae. (**F-F‴**) Representative trajectories of 7 dpf larvae with ablated (**F′**) *gsc2* (**F″**) *rln3a* NI or (**F‴**) *rln3a* PAG neurons and (**F**) sibling controls during the first 115 s of the recording (baseline activity). (**G**) Mean of distance traveled during the first 115 s of the recording for unablated controls (19.87±3.19 cm) or larvae with ablated *gsc2* (17.87±3.84193 cm), *rln3a* NI (42.80±5.27 cm), or *rln3a* PAG (15.73±3.55 cm) neurons. Kruskal-Wallis rank sum test: ***p=0.00099. Dunn's post-hoc tests with adjustment for multiple comparisons show ablated *rln3a* NI neurons vs. unablated **p=0.0019, ablated *rln3a* vs. *gsc2* NI neurons **p=0.0019, or ablated *rln3a* NI vs. *rln3a* PAG neurons **p=0.0019. (**H**) Average length of movement phases during the pre-shock period, defined as continuous phases of movement with no more than 1 s of prolonged immobility, for unablated controls (7.35±1.34 s) or larvae with ablated *gsc2* (6.18±1.14 s), *rln3a* NI (10.38±1.17 s) or *rln3a* PAG (4.23±0.74 s) neurons. Kruskal-Wallis rank sum test: **p=0.0013. Dunn's post-hoc tests with adjustment for multiple comparisons show ablated *rln3a* NI neurons vs. unablated *p=0.039, ablated *rln3a* vs. *gsc2* NI neurons *p=0.039, or ablated *rln3a* NI vs. *rln3a* PAG neurons ***p=0.00055. (**I**) Mean number of phases of movement during the pre-shock period for unablated controls (7.74±1.03) or larvae with ablated *gsc2* (6.88±1.39), *rln3a* NI (8.13±1.05), or *rln3a* PAG (7.41±1.33) neurons. Kruskal-Wallis rank sum test: p=0.89.

The online version of this article includes the following video and figure supplement(s) for figure 8:

**Figure supplement 1.** Confirmation of selective ablation of NI neuronal clusters.

**Figure supplement 2.** Loss of *rln3a* NI neurons increases turning behavior.

**Figure supplement 3.** Comparisons between ablated and unablated larvae with the same genotype.

**Figure 8—video 1.** Increased swimming behavior upon loss of *rln3a* NI neurons.

https://elifesciences.org/articles/89516/figures#fig8video1

A number of studies have found that aversive stimuli promote expression of c-Fos in the NI (*Lawther et al., 2015*; *Passerin et al., 2000*; *Rajkumar et al., 2016*; *Tanaka et al., 2005*), leading researchers to evaluate the role of relaxin-3 in anxiety-like behaviors. In rats, intracerebroventricular infusion of a relaxin-3 receptor agonist increases entries to the open arms of an elevated plus maze and the amount of time animals spend in the light portion of a light-dark box (*Ryan et al., 2013*). Similar assays in mice showed that the relaxin-3 receptor agonist did not alter the basal behavioral state but rather reduced anxiety-like behavior induced by the anxiogenic drug FG-7142 (*Zhang et al., 2015*). However, a role for the NI in regulating the behavioral response to acute aversive stimuli has so far not been described. *Lu et al., 2020* note that *Nmb* neurons in the mouse NI promote spontaneous locomotor activity and are activated in response to foot shock, a stimulus that elicits immediate locomotion, but whether they mediate the immediate locomotor response to this stimulus is unclear. Through selective ablation, we found that loss of either NI *rln3a* or *gsc2* neurons was not sufficient to alter hyperactivity normally observed in zebrafish larvae post-shock.

Previous work showed that zebrafish hindbrain *rln3a* neurons localize to a region expressing *corticotropin releasing hormone receptor 1* (*crhr1*), which encodes a receptor expressed at high levels in the rodent NI (*Bittencourt and Sawchenko, 2000*, *Potter et al., 1994*; ). We find that transcripts encoding neuromedin B and cholecystokinin, which have also been detected in the rodent NI (*Kubota et al., 1983*; *Lu et al., 2020*; *Olucha-Bordonau et al., 2003*), likewise map to the presumptive zebrafish NI. Similar to *rln3a* and *nmbb* neurons in the zebrafish larval NI, in mice *Rln3* and *Nmb* are expressed in

**Table 1.** Properties of *gsc2* and *rln3a* NI neurons.

| | Neurotransmitter identity | Projection pattern | Projections to IPN | Spontaneous activity | Influenced by dHb-IPN pathway | Locomotion post-ablation |
|---|---|---|---|---|---|---|
| *gsc2* neurons | 0% *slc17a6b⁺* 82.43±3.52% *gad1b⁺* | Widespread | Ventral IPN | Low spontaneous activity | Yes | No change |
| *rln3a* NI neurons | 0% *slc17a6b⁺* 80.57±5.57% *gad1b⁺* | Restricted to IPN | Dorsal IPN | Rhythmic calcium bursts | No | Increased |

interspersed neuronal populations and are co-expressed in a subset of cells (*Lu et al., 2020*; *Nasirova et al., 2020*). Furthermore, we found that *cckb* neurons are a separate population located posterior to the *rln3a* and *nmbb* neurons. *Szlaga et al., 2022* also found little overlap between cholecystokinin and relaxin-3 neurons in the rat brain. Together, these results suggest conservation of NI cell types and their organization from fish to mammals, establishing zebrafish as a model to understand the connectivity and function of the diverse types of NI neurons. Intriguingly, a new study identified a region in the zebrafish larval hindbrain, referred to as the dorsal tegmental nucleus, whose GABAergic neurons project to the dorsal IPN and are activated in conjunction with directional turning by the larva (*Petrucco et al., 2023*). Although the specific neuronal cell types have yet to be identified, it is likely they correspond to a subpopulation in the NI.

# Materials and methods

## Animals

Zebrafish were maintained at 27 °C under a 14:10 hr light/dark cycle in a recirculating system with dechlorinated, filtered and heated water (system water). All lines used are listed in the Appendix 1—key resources table. Larvae were screened for labeling by fluorescent proteins using an Olympus MVX10 Macro Zoom fluorescence microscope. For imaging, larvae were incubated in system water containing 0.003% phenylthiourea (P7629, Sigma-Aldrich) to inhibit melanin pigmentation. Most analyses were performed at the larval stage, before sex determination. Analyses performed at the adult stage included both males and females. All procedures were approved by the Institutional Animal Care and Use Committee (IACUC) of Dartmouth College (protocol #00002253).

## Generation of transgenic lines by Tol2 transposition

To generate *Tg(QUAS:GFP)$^{c578}$*, *Tg(QUAS:mApple, he1.1:CFP)$^{c788}$*, *Tg(QUAS:GFP-CAAX)$^{c591}$*, *Tg(QUAS:NLS-mApple, he1.1:CFP)$^{c718}$*, *Tg(QUAS:NLS-GFP; he1.1:CFP)$^{c682}$*, *Tg(QUAS:GFP-CAAX, he1.1:YFP)$^{c631}$* and *Tg(QUAS:GCaMP7a)$^{c594}$* transgenic lines, constructs for Tol2 transposition were produced using the MultiSite Gateway-based construction kit (*Kwan et al., 2007*). All plasmids used in this study and their Addgene identifiers are listed in the Appendix 1—key Resources Table. For each construct, three entry vectors were first assembled by BP reactions (11789020, Thermo Fisher Scientific). A 16 bp *QUAS* sequence (*Potter et al., 2010*) was cloned into the 5' entry vector (*pDON-RP4-P1R*, #219 of Tol2kit v1.2). DNA encoding GFP (green fluorescent protein) or mApple, or those sequences with an added nuclear localization sequence (NLS) or membrane localization sequence (CAAX) was inserted into middle entry vectors (*pDONR221*, #218 of Tol2kit v1.2). Sequences corresponding to the *SV40 poly A* tail, or the *poly A* tail followed by a secondary marker consisting of the zebrafish *hatching enzyme 1, tandem duplicate 1* (*he1.1*) promoter (*Xie et al., 2012*) driving CFP (cyan fluorescent protein) or YFP (yellow fluorescent protein), were placed into the 3' entry vector (*pDONR-P2R-P3*, #220 of Tol2kit v1.2). All three entry vectors were introduced into a Tol2 destination construct (*pDestTol2pA2*, #394 of the Tol2kit v1.2) using an LR reaction (11791020, Thermo Fisher Scientific).

To produce mRNA encoding Tol2 transposase, *pCS-zT2TP* (*Suster et al., 2009*) was digested with *Not*I and RNA was synthesized in vitro using the mMESSAGE mMACHINE Transcription Kit with SP6 polymerase (AM1340, Thermo Fisher Scientific). RNA was extracted with phenol/chloroform-isoamyl alcohol, re-extracted with chloroform, and precipitated with isopropanol. A solution containing *QUAS* plasmid DNA (25 ng/µl), Tol2 transposase mRNA (25 ng/µl) and phenol red (0.5%) was microinjected into one-cell stage zebrafish embryos that were raised to adulthood. Transgenic founders were identified by screening their F$_1$ progeny for fluorescently labeled hatching gland cells at 1 dpf and for labeling in the brain under QUAS control.

## Generation of transgenic lines by genome editing

Methods for CRISPR/Cas9-targeted integration were used to generate the *Tg(gsc2:QF2)$^{c721}$* and *Tg(rln3a:QF2, he1.1:YFP)$^{c836}$* driver lines. For *Tg(gsc2:QF2)$^{c721}$*, the non-homologous end joining technique described by *Kimura et al., 2014* was modified by integration of a *QF2* donor plasmid, *Gbait-hsp70-QF2-pA* (Addgene plasmid #122563), which contains a GFP bait sequence for Cas9-mediated linearization of the donor plasmid (*Kimura et al., 2014*). Cas9 RNA (*Jao et al., 2013*) and sgRNAs (*Hwang et al., 2013*) targeting *gsc2* or the GFP bait sequence (*Auer et al., 2014*) were synthesized as previously

described. Briefly, pairs of synthetic oligonucleotides (*gsc2*_sense, *gsc2*_anti-sense, Appendix 1—key Resources Table), containing the overhangs 5'-TAGG-N$_{18}$-3' (sense) or 5'-AAAC-N$_{18}$-3' (anti-sense), were annealed to each other. The resulting DNA was cloned into the *pDR274* vector Addgene, plasmid #42250; *Hwang et al., 2013* following digestion of *pDR274* with *Bsa*I (R3733S, New England Biolabs). The *pDR274* templates and the *pDR274* vector for synthesis of the GFP bait sgRNA (*Auer et al., 2014*) were digested by *Dra*I and sgRNAs synthesized using the MAXIscript T7 Transcription Kit (AM1312, Thermo Fisher Scientific). *pT3TS-nCas9n* template DNA (Addgene, plasmid #46757; *Jao et al., 2013*) was digested with *Xba*I (R0145S, New England Biolabs), and Cas9 RNA was synthesized using the mMESSAGE mMACHINE Transcription Kit (AM1348, Thermo Fisher Scientific). For each transgenic line, a solution containing the sgRNA targeting the gene of interest (50 ng/µl), GFP bait sgRNA (50 ng/µl), the *Gbait-hsp70-QF2-pA* plasmid (50 ng/µl), Cas9 mRNA (500 ng/µl), and phenol red (0.5%) was microinjected into one-cell stage embryos.

For *Tg(rln3a:QF2, he1.1:YFP)$^{c836}$*, the GeneWeld approach described by Wierson et al., which uses short homology arms to facilitate integration by homology-directed repair, was modified by introduction of *QF2* and *he1.1:YFP* sequences into the donor vector (*Wierson et al., 2020*). The resulting *pPRISM-QF2-he1.1:YFP* donor construct contains two target sites for a universal sgRNA (ugRNA) that flank the cargo: a 2 A self-cleaving sequence, *QF2*, and the *he1.1:YFP* secondary marker. To generate the construct, four PCR products were produced. *QF2* was amplified from *Gbait-hsp70-QF2-pA* (Addgene plasmid #122563; *Choi et al., 2021*; _QF2_F, 2 A_QF2_R, Appendix 1—key Resources Table). The *he1.1:YFP* cassette (*he1.1:YFP_F*, *he1.1:YFP_R*, Appendix 1—key Resources Table) was amplified from *p3E_he1a:YFP* (Addgene, plasmid #113879), and the polyA terminator (polyA_F, polyA_R, Appendix 1—key Resources Table) and plasmid backbone (Col1E_F, Col1E_R, Appendix 1—key Resources Table) were amplified from *pPRISM-Stop-cmlc2-eGFP* (Addgene kit #1000000154; *Wierson et al., 2020*). The PCR-amplified fragments were assembled using NEBuilder HiFi DNA Assembly Cloning Kit (E5520S, New England Biosystems).

To produce *rln3a* homology arms, complementary oligonucleotide pairs (*rln3a*_5'arm_sense, *rln3a*_5'arm_anti-sense; *rln3a*_3'arm_sense, *rln3a*_3'arm_anti-sense, Appendix 1—key Resources Table) were designed using GTagHD (*Wierson et al., 2020*) and annealed to each other. The *pPRISM-QF2-he1.1:YFP* donor vector was first digested with *Bfu*AI and *Bsp*QI, (R0701S and R0712S, New England Biolabs) and then combined with the homology arms in a ligation reaction (M0202S, New England Biolabs). To synthesize ugRNA and an sgRNA targeting the *rln3a* gene, synthetic oligonucleotide pairs (*rln3a*_sense, ugRNA_sense, common_anti-sense, Appendix 1—key Resources Table) were annealed to each other, elongated by Phusion polymerase (M0530S, New England Biolabs), and used as templates for in vitro transcription with the MAXIscript T7 Transcription Kit (AM1312, Thermo Fisher Scientific). A solution containing *rln3a* sgRNA (50 ng/µl), universal sgRNA (50 ng/µl), the *pPRISM-QF2-he1.1:YFP-rln3a-HA* donor plasmid (100 ng/µl), Cas9 mRNA (500 ng/µl), and phenol red (0.5%) was microinjected into one-cell stage embryos.

When applicable, injected embryos were screened for labeling by fluorescent proteins in the hatching gland. To verify successful integration, PCR was performed on genomic DNA from injected embryos using primers that flank the integration site, with the forward primer corresponding to genomic sequence and the reverse primer corresponding to donor plasmid sequence (*gsc2*_val_F, hsp70_R; *rln3a*_val_F, QF2_R, Appendix 1—key Resources Table). Sanger sequencing confirmed the identity of PCR products. Transgenic founders were identified by breeding F$_0$ adults with a *QUAS* reporter line and screening progeny for fluorescent labeling of the hatching gland when applicable, and for labeling by QUAS-driven fluorescent reporters. PCR and sequencing were repeated in F$_1$ larvae to confirm integration at the correct target site.

## RNA in situ hybridization

DNA templates for *gsc2*, *rln3a*, *ccka*, *cckb* probes were generated using PCR to incorporate a binding site for SP6 polymerase. cDNA for PCR amplification was obtained by reverse transcription of RNA extracted from 6 dpf embryos with TRIzol (15596026, Invitrogen) using the QuantiTect Reverse Transcription kit (205311, QIAGEN). DNA templates were amplified with the following PCR primers: *gsc2*_F, *gsc2*_R, *rln3a*_F, *rln3a*_R, *ccka*_F, *ccka*_R, *cckb*_F, and *cckb*_R (Appendix 1—key Resources Table). DNA templates for *nmba*, *nmbb* and *nts* were amplified from cDNA (*nmba*_F, *nmba*_R, *nmbb*_F, *nmbb*_R, *nts*_F, *nts*_R, Appendix 1—key Resources Table), cloned using the TOPO TA kit (K465001, Invitrogen),

and linearized by digestion with *BamH*I (R0136S, New England Biolabs). The template for the *sst1.1* probe was a cDNA clone in a *pSPORT1* vector (*Argenton et al., 1999*) linearized by digestion with *Sal*I (R3138L, New England Biolabs).

DNA templates were used for digoxigenin (DIG)-labeled in vitro transcription of *gsc2*, *rln3a*, *ccka*, *cckb*, *nmba*, *nmbb,* and *nts* probes (11175025910, Roche) and fluorescein (FITC)-labeled in vitro transcription of *rln3a* and *sst1.1* probes (11685619910, Roche). The *gsc2*, *rln3a*, *ccka*, *cckb*, and *sst1.1* probes were synthesized with SP6 polymerase and the *nmba*, *nmbb*, and *nts* probes with T7 polymerase (Fisher Scientific, EP0113). RNA probes were purified using illustra MicroSpin G-50 Columns (27533001, GE Healthcare).

For whole-mount RNA in situ hybridization ( *Liang et al., 2000*; *Thisse et al., 1993*), larvae and dissected adult brains were fixed overnight in paraformaldehyde (PFA; 4% in 1 x phosphate-buffered saline) at 4 °C then dehydrated overnight in 100% methanol (A4124, Fisher Scientific) at –20 °C. Tissue was rehydrated stepwise in methanol/phosphate-buffered saline (PBS) and washed with PBT (1 x PBS, 0.1% Tween 20). Larvae were digested for 30 min and dissected adult brains for 35 min in proteinase K (3115836001, Roche; 10 µg/ml in PBT). To stop the reaction, tissue was fixed in 4% PFA at room temperature for 20 min, then washed with PBT. Specimens were prehybridized for at least two hours at 70 °C in hybridization buffer [50% formamide (17899, Fisher Scientific), 5 X saline sodium citrate (SSC), 50 µg/ml heparin (H3393, Sigma-Aldrich), 500 µg/ml tRNA (10109525001, Sigma-Aldrich), 0.1% Tween 20 (P1379, Sigma-Aldrich), 9 mM citric acid] with 5% dextran and then hybridized overnight at 70 °C in hybridization buffer with 5% dextran and 30 ng of probe. Samples were then washed in hybridization buffer (without dextran), transitioned stepwise at 70 °C from hybridization buffer to 2 X SSC, washed twice for 30 minutes in 0.2 X SSC at 70 °C, and transitioned stepwise into PBT at room temperature. Adult brains were embedded in 4% low melting point agarose (50100, Lonza) and sectioned (70 µm) using a Leica VT1000s vibratome. Whole mount larvae and adult brain sections layered on glass slides were blocked for at least one hour in PBT with 2 mg/ml bovine serum albumin and 2% sheep serum at room temperature and then incubated overnight at 4 °C with alkaline phosphatase-coupled anti-DIG antiserum (11093274910, Roche) diluted 1/5000 in blocking solution. Samples were washed several times in PBT, and detection with 4-Nitro blue tetrazolium chloride (NBT; 11383213001, Roche) and 5-bromo-4-chloro-3-indolyl-phosphate (BCIP; 11383221001, Roche) was performed in alkaline phosphatase reaction buffer (100 mM Tris pH 9.5, 50 mM MgCl$_2$, 100 mM NaCl, 0.1% Tween 20).

For colorimetric double in situ hybridization reactions, larvae were hybridized with DIG and FITC probes simultaneously as previously described (*Liang et al., 2000*), and the DIG probe was first detected using NBT/BCIP as above. To inactivate alkaline phosphatase, larvae were post-fixed overnight at room temperature in 4% PFA, washed twice for 20 min each with MABT (100 mM maleic acid, 150 mM NaCl, 0.1% Tween-20, pH 7.5), incubated for 10 min at 70 °C in EDTA (10 mM in MABT), and dehydrated in methanol for 10 min. Samples were rehydrated stepwise in methanol/MABT, washed in MABT, and blocked for 1 hr in blocking buffer consisting of 20% sheep serum and 2% blocking reagent (11096176001, Roche) in MABT. Tissues were incubated overnight at 4 °C in alkaline phosphatase-coupled anti-FITC antiserum (11426338910, Roche) diluted 1:5000 in blocking buffer. Finally, samples were washed several times in MABT. FITC detection with BCIP and iodo-nitrotetrazolium violet was performed in alkaline phosphatase buffer with 10% polyvinyl alcohol. Samples were cleared in glycerol and mounted for imaging using a Zeiss Axioskop microscope fitted with a Leica DFC 500 digital color camera and Leica Applications Suite software.

For fluorescent double in situ hybridization, larvae were fixed in 4% PFA, dehydrated in methanol, and incubated in 2% hydrogen peroxide in methanol for 20 min. After rehydration and washing in PBT as above, larvae were digested for 30 min in 20 µg/ml proteinase K in PBT, post-fixed in 4% PFA, washed, prehybridized, and hybridized overnight at 70 °C in hybridization buffer with 5% dextran and 40 ng each of DIG and FITC probes. Stringency washes were performed as above, then larvae were washed in TNT [0.1 M Tris pH 7.5, 0.1 M NaCl, 0.1% Tween-20] and maintained for 2 hr in 2% blocking reagent (11096176001, Roche) in TNT. Larvae were incubated overnight at 4 °C in horseradish peroxidase-coupled anti-FITC antiserum (11426346910, Roche) diluted 1:500 in blocking solution, then washed several times in TNT. FITC detection was performed using TSA Plus fluorescein diluted 1:50 in amplification diluent (NEL741001KT, Akoya Biosciences). Samples were washed several times in TNT, incubated in 1% hydrogen peroxide in TNT for 20 min, washed again in TNT, blocked as above for 1 hr, and incubated overnight at 4 °C in horseradish peroxidase-coupled anti-DIG antiserum

(11207733910, Roche) diluted 1:500 in blocking solution. Tissue was washed several more times in TNT and DIG detection was performed using TSA Plus Cyanine diluted 1:50 in amplification diluent (NEL744001KT, Akoya Biosciences). Fluorescently labeled samples were imaged using confocal microscopy.

## Confocal imaging

Larvae were anesthetized in 0.02% tricaine and individually mounted in a droplet of 1.5% low melting point agarose (50100, Lonza) centered in a 60 mm x 15 mm Petri dish. After the agarose solidified, system water with 0.02% tricaine was added to each dish. Larvae were imaged using either a Leica SP5 with a 25 X (NA=0.95) water immersion objective, or a Zeiss LSM 980 with a 20 X (NA=0.5) water immersion objective.

Adult brains were fixed overnight in 4% PFA at 4 °C, rinsed in 1 X PBS, and mounted in 4% low melting point agarose (50100, Lonza) for sectioning (70 µm) by a Leica VT1000s vibratome. Sections were mounted in glycerol for imaging under either a Leica SP5 with a 20 X (NA=0.7) objective or a Zeiss LSM 980 with a 20 X (NA=0.8) objective.

Z-stacks of the larval brain encompassing fluorescent signals included approximately 125 slices and 250 µm for dorsal views, or 75 slices and 150 µm for lateral views. Z-stacks focused only on the NI included approximately 35 slices and 70 µm from a dorsal or lateral view. Z-stacks of adult brain sections included 35 slices and 70 µm.

## Calcium signaling

Larvae were paralyzed by a 1 min immersion in α-bungarotoxin (20 µl of 1 mg/ml solution in system water; B1601, Thermo Fisher Scientific), followed by washing in fresh system water (*Baraban, 2013*; *Duboué et al., 2017*; *Severi et al., 2014*). Individual larvae were embedded in a droplet of 1.5% low melting point agarose (50100, Lonza) centered in a 60 mm x 15 mm Petri dish. After the agarose solidified, system water was added to the dish. For all calcium signaling experiments, larvae were imaged in *xyt* acquisition mode using a Zeiss LSM 980 with a 20 X (NA=0.5) water immersion objective and a 488 nm laser.

To record calcium transients in response to electric shock, a plastic ring holding electrodes that were connected to a Grass SD9 electrical stimulator (Grass Instruments), was placed in each dish. Images of *gsc2* or *rln3a* NI neurons were acquired at 475 x 475 pixel resolution and a rate of 5.2 Hz. Calcium transients were recorded for 600 frames (115.4 s) for baseline measurements, then larvae were shocked once (25 V, 200 ms duration) and 1400–1800 more frames (269.2–346.2 s) collected.

To record calcium transients in response to stimulation of the red-shifted opsin ReaChR (*Lin et al., 2013*) with 561 nm light, images were first acquired using a 488 nm laser at 310 x 310 pixel resolution and a rate of 2.6 Hz. The Z-depth was adjusted to the plane of the neuronal population being imaged (i.e. dHb, NI, or PAG brain regions). Spontaneous calcium transients were recorded for 200 frames (76.9 s), the 561 nm laser was activated at 5% power while 20 more frames (7.7 s) were acquired, and then calcium transients were recorded for another 150 frames (57.7 s).

For all calcium imaging experiments, individual frames were extracted in Fiji (*Schindelin et al., 2012*) using *File ->Save As ->Image Sequence* and imported to MATLAB, where mean fluorescence intensities for regions of interest (ROI) were calculated. Briefly, a high contrast image was generated for each larva by calculating a maximum intensity projection of its image series. ROIs were drawn manually using the high contrast image and the MATLAB function *roipoly*. For recordings of *gsc2* or *rln3a* neurons, ROIs were individual neurons; for dHb recordings, each dHb nucleus was designated as an ROI. Mean fluorescence intensity of pixels within each ROI was calculated. ΔF/F was calculated according to the following formula:

$$F \leftarrow \frac{F_i - F_{min}}{F_{max} - F_{min}}$$

where $F_i$ indicates the mean fluorescence intensity in an ROI at each time point, and $F_{max}$ and $F_{min}$ are the maximum and minimum fluorescence values respectively for that ROI during the recording period. To calculate total activity before and after the stimulus, ΔF/F was averaged across ROIs for each larva and the total activity was obtained for the time period by calculating the area under the curve using the MATLAB function *trapz*.

The initial time point at which neuronal activity increased for a given ROI was calculated using the MATLAB *findpeaks* function (MinPeakProminence: 0.2, MinPeakWidth: 10).

## Two-photon laser-mediated cell ablation

At 6 dpf, *Tg(gsc2:QF2)*[c721]; *Tg(QUAS:GFP)*[c578] or *Tg(rln3a:QF2, he1.1:YFP)*[c836]; *Tg(QUAS:GFP)*[c578] larvae were anesthetized in 0.02% tricaine and individually mounted within a droplet of 1.5% low melting point agarose (50100, Lonza) centered in a 30 mm x 10 mm Petri dish. After the agarose solidified, system water was added. GFP-expressing cells were located using a two-photon microscope (Bruker) with a 60 X (NA=1) objective. The laser was tuned to 885 nm and, using GFP labeling as a guide, was focused on the relevant neuronal population and activated for several seconds at maximum power until the GFP signal disappeared. Because the two-photon laser power is delivered to a restricted Z-plane, ablations were repeated at multiple depths to eliminate each cell population. For ablation of *gsc2* neurons the laser was activated over an area of 600–2000 µm² on four Z-planes. The laser was activated over an area of 1000–1250 µm² on two Z-planes for ablation of *rln3a* NI neurons and over an area of 1200–1800 µm² on two Z-planes for removal of each *rln3a* PAG nucleus (left and right).

## Locomotor assay

Behavioral experiments were performed blind to the ablation status of each larva being assayed. Unablated controls were a mixture of *Tg(gsc2:QF2)*[c721]; *Tg(QUAS:GFP)*[c578] and *Tg(rln3a:QF2, he1.1:Y-FP)*[c836]; *Tg(QUAS:GFP)*[c578], and were siblings of ablated larvae. Behavioral tests were conducted in a temperature-controlled room (27 ° C) on individual 7 dpf larvae. The 6 cm³ acrylic testing chamber had a 0.5 cm platform on which a 40 mm cell strainer (Falcon) was placed. The chamber was filled with fresh system water and set on top of an infrared illumination source (880 nm, ViewPoint Life Sciences). Locomotor activity was recorded by a high frame rate charged-coupled device (CCD) camera (Point Grey Research), which was connected to a computer (Dell). Tracking was performed in real time at 60 frames per second, using ZebraLab software (ViewPoint Life Sciences). Swimming behavior was recorded for 120 s, then each larva was shocked once (25 V, 200ms duration), and activity recorded for an additional 120 s. To analyze locomotor activity, the x and y coordinates of a larva's position in each frame were exported from ZebraLab. Activity was quantified using R statistical software (*R Development Core Team, 2023*) according to the following equation:

$$D = \sqrt{\left(x_{i+1} - x_i\right)^2 + \left(y_{i+1} - y_i\right)^2}$$

where i indicates a single frame. Total distances were calculated by summing the distance for all frames over the relevant period of the recording. Total number of movement phases and average length of movement phases during the pre-shock period were calculated for each larva using R statistical software (*R Development Core Team, 2023*). Movement trajectories were plotted using MATLAB. Phases of movement and their durations were extracted using R statistical software by first binning the distances for all frames within each second of the recording, then iterating through each subsequent second. A movement phase was determined to commence when there was movement within a given second and no movement in the preceding second, and the phase persisted for each subsequent consecutive second with movement. The phase of movement was determined to end when there was no movement within a given second following one with movement. Turning behavior was quantified using R statistical software by iterating through the x and y coordinates of a larva's position in each frame. Starting with the second frame, the size of the angle formed by the larva's change in orientation between the first and second frame, and the second and third frame, was calculated according to the law of cosines:

$$C = \cos^{-1}\left(\left(a^2 + b^2 - c^2\right)/2ab\right)$$

where C is the angle formed by two lines of length a and b, and c is the length of the side opposite angle C. Total turning was calculated by summing the size, in degrees, of all calculated angles of all frames in which a larva was active. Total turning for each larva was divided by the total distance traveled to calculate a ratio of turning per unit of distance traveled.

**Table 2.** Summary of statistical tests used.

| Figure | Panel | Data structure | Type of test | p value |
|---|---|---|---|---|
| 6 | D″ | Normal | Two-sample t-test | p=0.00041 |
| 6 | E″ | Normal | Two-sample t-test | p=0.0073 |
| 6 | F″ | Non-parametric | Wilcoxon rank sum test | p=0.032 |
| 6 | G″ | Normal | Two-sample t-test | p=0.83 |
| 6 | H″ | Normal | Two-sample t-test | p=0.45 |
| 7 | H | Normal | Two-sample t-test | p=0.0035 |
| 7 | I | Non-parametric | Wilcoxon rank sum test | p=0.04 |
| 8 | E | Non-parametric | Kruskal-Wallis rank-sum test with Dunn's all-pairs test | Kruskal-Wallis: p=2.2 x 10$^{-16}$. Dunn's post-hoc tests: no statistically significant differences within pre-shock and post-shock groups, p<0.001 for each pre-shock vs. post-shock comparison. |
| 8 | G | Non-parametric | Kruskal-Wallis rank-sum test with Dunn's all-pairs test | Kruskal-Wallis: p=0.00099. *rln3a* NI neurons ablated vs. unablated p=0.0019 *rln3a* NI neurons ablated vs. *gsc2* neurons ablated p=0.0019 *rln3a* NI neurons ablated vs. *rln3a* PAG neurons ablated p=0.0019. |
| 8 | H | Non-parametric | Kruskal-Wallis rank-sum test with Dunn's all-pairs test | Kruskal-Wallis: p=0.0013. *rln3a* NI neurons ablated vs. unablated p=0.039, *rln3a* NI neurons ablated vs. *gsc2* neurons ablated p=0.039, *rln3a* NI neurons ablated vs. *rln3a* PAG neurons ablated p=0.00055. |
| 8 | I | Non-parametric | Kruskal-Wallis rank-sum test with Dunn's all-pairs test | *p*=0.89 |
| 8–2 | A | Non-parametric | Kruskal-Wallis rank-sum test with Dunn's all-pairs test | Kruskal-Wallis: p=0.00017. ablated *rln3a* NI neurons vs. unablated p=0.00038, ablated *rln3a* vs. *gsc2* NI neurons p=0.00066, or ablated *rln3a* NI vs. *rln3a* PAG neurons p=0.00066. |
| 8–2 | B | Non-parametric | Kruskal-Wallis rank-sum test with Dunn's all-pairs test | Kruskal-Wallis: p=0.000097. ablated *rln3a* NI neurons vs. unablated p=0.00057, ablated *rln3a* vs. *gsc2* NI neurons p=0.00059, or ablated *rln3a* NI vs. *rln3a* PAG neurons p=0.00061. |
| 8–2 | C | Non-parametric | Kruskal-Wallis rank-sum test with Dunn's all-pairs test | Kruskal-Wallis: p=0.00045. ablated *rln3a* NI neurons vs. unablated p=0.0045, ablated *rln3a* vs. *gsc2* NI neurons p=0.001, or ablated *rln3a* NI vs. *rln3a* PAG neurons p=0.001. |
| 8–3 | A | Non-parametric | Kruskal-Wallis rank-sum test with Dunn's all-pairs test | Kruskal-Wallis: p=4.74 x 10$^{-15}$. Dunn's post-hoc tests with adjustment for multiple comparisons: no statistically significant differences within pre-shock and post-shock groups, p<0.001 for each pre-shock vs. post-shock comparison. |
| 8–3 | B | Non-parametric | Kruskal-Wallis rank-sum test with Dunn's all-pairs test | Kruskal-Wallis: p=2.85 x 10$^{-16}$. Dunn's post-hoc tests with adjustment for multiple comparisons: no statistically significant differences within pre-shock and post-shock groups, p<0.001 for each pre-shock vs. post-shock comparison. |
| 8–3 | C | Non-parametric | Kruskal-Wallis rank-sum test with Dunn's all-pairs test | Kruskal-Wallis: p=0.0012. ablated *rln3a* NI neurons vs. unablated p=0.012, ablated *rln3a* vs. *gsc2* NI neurons p=0.0023, or ablated *rln3a* NI vs. *rln3a* PAG neurons p=0.0023. |
| 8–3 | D | Non-parametric | Kruskal-Wallis rank-sum test with Dunn's all-pairs test | Kruskal-Wallis: p=0.0018. ablated *rln3a* NI neurons vs. unablated p=0.098, ablated *rln3a* vs. *gsc2* NI neurons ablated p=0.060, or ablated *rln3a* NI vs. *rln3a* PAG neurons p=0.00076. |
| 8–3 | E | Non-parametric | Kruskal-Wallis rank-sum test with Dunn's all-pairs test | p=0.87 |
| 8–3 | F | Non-parametric | Kruskal-Wallis rank-sum test with Dunn's all-pairs test | Kruskal-Wallis: p=0.0012. ablated *rln3a* NI neurons vs. unablated p=0.0078, ablated *rln3a* vs. *gsc2* NI neurons p=0.0022, or ablated *rln3a* NI vs. *rln3a* PAG neurons p=0.0022. |
| 8–3 | G | Non-parametric | Kruskal-Wallis rank-sum test with Dunn's all-pairs test | Kruskal-Wallis: p=0.0012. ablated *rln3a* NI neurons vs. unablated p=0.099, ablated *rln3a* vs. *gsc2* NI neurons ablated p=0.042, or ablated *rln3a* NI vs. *rln3a* PAG neurons p=0.00049. |
| 8–3 | H | Non-parametric | Kruskal-Wallis rank-sum test with Dunn's all-pairs test | p=0.90 |

## Quantification and statistical analyses

All means are presented with standard error of the mean. Statistical details for all experiments are summarized in *Table 2*. Data structure was determined using Shapiro-Wilk tests. Statistical analyses were performed using either R statistical software (*R Development Core Team, 2023*) or MATLAB. Sample sizes were similar to those typically used in zebrafish behavior and calcium imaging studies (*Agetsuma et al., 2010*; *Choi et al., 2021*; *Facchin et al., 2015*; *Muto et al., 2013*; *Wee et al., 2019*). Data were plotted using the MATLAB library PlotPub (*Habib Masum, 2022*) or the R package ggplot2 (*Wickham, 2016*). Where applicable, larvae were randomized to the treatment or control group, and no larvae were excluded.

## Acknowledgements

We thank Dr. Bryan Luikart for sharing his expertise and equipment for two-photon microscopy, Jean-Michael Chanchu for generating *QUAS* transgenic lines, Essence Vinson and Ming Wu for assistance with RNA in situ hybridization, Dr. Jeffrey Mumm and Dr. Filippo Del Bene for providing plasmids, and Dr. Rejji Kuruvilla and Dr. Erik Duboué for valuable feedback on the manuscript.

## Additional information

### Funding

| Funder | Grant reference number | Author |
|---|---|---|
| National Institutes of Health | R37HD091280 | Marnie E Halpern |
| National Science Foundation | DGE-1746891 | Emma D Spikol |

The funders had no role in study design, data collection and interpretation, or the decision to submit the work for publication.

### Author contributions

Emma D Spikol, Conceptualization, Data curation, Software, Formal analysis, Funding acquisition, Validation, Investigation, Visualization, Methodology, Writing – original draft; Ji Cheng, Methodology, Writing – review and editing; Michelle Macurak, Methodology; Abhignya Subedi, Investigation; Marnie E Halpern, Conceptualization, Resources, Supervision, Funding acquisition, Project administration, Writing – review and editing

### Author ORCIDs

Emma D Spikol http://orcid.org/0000-0002-0565-1537
Ji Cheng http://orcid.org/0000-0003-1610-2557
Marnie E Halpern https://orcid.org/0000-0002-3634-9058

### Ethics

All procedures were approved by the Institutional Animal Care and Use Committee (IACUC) of Dartmouth College (protocol #00002253).

Reviewer #1 (Public Review): https://doi.org/10.7554/eLife.89516.3.sa1
Reviewer #3 (Public Review): https://doi.org/10.7554/eLife.89516.3.sa2
Reviewer #4 (Public Review): https://doi.org/10.7554/eLife.89516.3.sa3
Author response https://doi.org/10.7554/eLife.89516.3.sa4

## Additional files

### Supplementary files
• MDAR checklist

### Data availability

Raw data have been deposited at Mendeley Data and are publicly available. All original code has been deposited at Zenodo and is publicly available. DOIs for data and code are listed in the Key Resources Table.

The following datasets were generated:

| Author(s) | Year | Dataset title | Dataset URL | Database and Identifier |
|-----------|------|---------------|-------------|-------------------------|
| Spikol ED, Cheng J, Macurak M, Subedi A, Halpern ME | 2024 | Genetically Defined Nucleus Incertus Neurons Differ in Connectivity and Function--Spikol et al--part 1 | http://doi.org/10.17632/tm2bjzjp5g.1 | Mendeley Data, 10.17632/tm2bjzjp5g.1 |
| Spikol ED, Cheng J, Macurak M, Subedi A, Halpern ME | 2024 | Genetically Defined Nucleus Incertus Neurons Differ in Connectivity and Function--Spikol et al--part 2 | http://doi.org/10.17632/mcbdr53ppt.1 | Mendeley Data, 10.17632/mcbdr53ppt.1 |
| Spikol ED, Cheng J, Macurak M, Subedi A, Halpern ME | 2024 | Genetically Defined Nucleus Incertus Neurons Differ in Connectivity and Function--Spikol et al--part 3 | http://doi.org/10.17632/3vrhjh6xrp.1 | Mendeley Data, 10.17632/3vrhjh6xrp.1 |
| Spikol ED, Cheng J, Macurak M, Subedi A, Halpern ME | 2024 | Genetically Defined Nucleus Incertus Neurons Differ in Connectivity and Function--Spikol et al--part 4 | http://doi.org/10.17632/p9nd6mf7w2.1 | Mendeley Data, 10.17632/p9nd6mf7w2.1 |
| Spikol ED, Cheng J, Macurak M, Subedi A, Halpern ME | 2024 | Genetically Defined Nucleus Incertus Neurons Differ in Connectivity and Function--Spikol et al--part 5 | http://doi.org/10.17632/pmpxtfv2ps.1 | Mendeley Data, 10.17632/pmpxtfv2ps.1 |
| Spikol ED, Cheng J, Macurak M, Subedi A, Halpern ME | 2024 | Genetically Defined Nucleus Incertus Neurons Differ in Connectivity and Function--Spikol et al--part 6 | http://doi.org/10.17632/xwtjpvd885.1 | Mendeley Data, 10.17632/xwtjpvd885.1 |

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

# Appendix 1

## Appendix 1—key resources table

| Reagent type (species) or resource | Designation | Source or reference | Identifiers | Additional information |
|---|---|---|---|---|
| Gene (*Danio rerio*) | gsc2 | Ensembl (Zv9) | ENSDARG00000076491 | |
| Gene (*Danio rerio*) | rln3a | Ensembl (GRCz11) | ENSDARG00000070780 | |
| Gene (*Danio rerio*) | ccka | Ensembl (GRCz11) | ENSDARG00000070810 | |
| Gene (*Danio rerio*) | cckb | Ensembl (GRCz11) | ENSDARG00000100052 | |
| Gene (*Danio rerio*) | nmba | Ensembl (GRCz11) | ENSDARG00000068144 | |
| Gene (*Danio rerio*) | nmbb | Ensembl (GRCz11) | ENSDARG00000077167 | |
| Gene (*Danio rerio*) | nts | Ensembl (GRCz11) | ENSDARG00000057887 | |
| Gene (*Danio rerio*) | sst1.1 | Ensembl (GRCz11) | ENSDARG00000040799 | |
| Strain (*Danio rerio*) | AB | *Walker, 1999* | RRID: ZIRC_ZL1 | |
| Genetic reagent (*Danio rerio*) | TgBAC(gng8:Eco.NfsB-2A-CAAX-GFP)$^{c375}$ | *deCarvalho et al., 2013* | RRID: ZFIN_ZDB-GENO-130815-4 | |
| Genetic reagent (*Danio rerio*) | Tg(gsc2:QF2)$^{c721}$ | This manuscript | N/A | Available by request from Halpern Lab |
| Genetic reagent (*Danio rerio*) | Tg(rln3a:QF2, he1.1:YFP)$^{c836}$ | This manuscript | N/A | Available by request from Halpern Lab |
| Genetic reagent (*Danio rerio*) | Tg(QUAS:GFP)$^{c578}$ | This manuscript | N/A | Available by request from Halpern Lab |
| Genetic reagent (*Danio rerio*) | Tg(slc17a6b:DsRed)$^{nns9}$ | *Miyasaka et al., 2009* | RRID: ZFIN_ZDB-GENO-100505-14 | |
| Genetic reagent (*Danio rerio*) | Tg(QUAS:mApple, he1.1:CFP)$^{c788}$ | This manuscript | N/A | Available by request from Halpern Lab |
| Genetic reagent (*Danio rerio*) | Tg(slc17a6b:EGFP)$^{zf139Tg}$ | *Miyasaka et al., 2009* | RRID: ZFIN_ZDB-GENO-090716-2 | |
| Genetic reagent (*Danio rerio*) | Tg(QUAS:mApple-CAAX, he1.1:mCherry)$^{c636}$ | This manuscript | N/A | Available by request from Halpern Lab |
| Genetic reagent (*Danio rerio*) | TgBac(gad1b:GFP)$^{nns25}$ | *Satou et al., 2013* | RRID: ZFIN_ZDB-GENO-131127-6 | |
| Genetic reagent (*Danio rerio*) | Tg(QUAS:GFP-CAAX)$^{c591}$ | This manuscript | N/A | Available by request from Halpern Lab |

*Appendix 1 Continued on next page*

*Appendix 1 Continued*

| Reagent type (species) or resource | Designation | Source or reference | Identifiers | Additional information |
|---|---|---|---|---|
| Genetic reagent (*Danio rerio*) | TgBAC(gng8:GAL4FF)*c426* | *Hong et al., 2013* | RRID: ZFIN_ZDB-GENO-140423-3 | |
| Genetic reagent (*Danio rerio*) | Tg(UAS-E1B:NTR-mCherry)*c264* | *Davison et al., 2007* | RRID: ZFIN_ZDB-GENO-070316-1 | |
| Genetic reagent (*Danio rerio*) | Tg(QUAS:NLS-mApple, he1.1:CFP)*c718* | This manuscript | N/A | Available by request from Halpern Lab |
| Genetic reagent (*Danio rerio*) | Tg(QUAS:NLS-GFP, he1.1:CFP)*c682* | This manuscript | N/A | Available by request from Halpern Lab |
| Genetic reagent (*Danio rerio*) | Tg(QUAS:GFP-CAAX, he1.1:YFP)*c631* | This manuscript | N/A | Available by request from Halpern Lab |
| Genetic reagent (*Danio rerio*) | Tg(QUAS:GCaMP7a)*c594* | This manuscript | N/A | Available by request from Halpern Lab |
| Genetic reagent (*Danio rerio*) | Tg(UAS:GCaMP7a)*zf415* | *Muto et al., 2013* | RRID:ZFIN_ZDB-GENO-131120-53 | |
| Genetic reagent (*Danio rerio*) | Tg(UAS:ReaChR-RFP)*jf50* | *Wee et al., 2019* | ZDB-ALT-201105–3 | |
| recombinant DNA reagent | pDestTol2-QUAS:GFP | This manuscript | Addgene plasmid #184811 | Available from Addgene |
| Recombinant DNA reagent | pDestTol2-QUAS:mApple-he1.1:CFP | This manuscript | Addgene plasmid #184812 | Available from Addgene |
| Recombinant DNA reagent | pDestTol2-QUAS:GFP-CAAX | This manuscript | Addgene plasmid #184813 | Available from Addgene |
| Recombinant DNA reagent | pDestTol2-QUAS:NLS-mApple-he1.1:CFP | This manuscript | Addgene plasmid #184814 | Available from Addgene |
| Recombinant DNA reagent | pDestTol2-QUAS:NLS-GFP-he1.1:CFP | This manuscript | Addgene plasmid #184815 | Available from Addgene |
| Recombinant DNA reagent | pDestTol2-QUAS:GFP-CAAX-he1.1:YFP | This manuscript | Addgene plasmid #184816 | Available from Addgene |
| Recombinant DNA reagent | pDestTol2-QUAS:GCaMP7a | This manuscript | Addgene plasmid #184817 | Available from Addgene |
| Recombinant DNA reagent | pCS-zT2TP | *Suster et al., 2009* | N/A | |
| Recombinant DNA reagent | Gbait-hsp70-QF2-pA | *Choi et al., 2021* | Addgene plasmid #122563 | |
| Recombinant DNA reagent | pDR274 | *Hwang et al., 2013* | Addgene plasmid #42250 | |
| Recombinant DNA reagent | pDR274-gsc2-sgRNA | This manuscript | Addgene plasmid #184818 | Available from Addgene |
| Recombinant DNA reagent | pDR274-GFPbait-sgRNA | *Auer et al., 2014* | N/A | |
| Recombinant DNA reagent | pT3TS-nCas9n | *Jao et al., 2013* | Addgene plasmid #46757 | |
| Recombinant DNA reagent | pPRISM-QF2-he1.1:YFP | This manuscript | Addgene plasmid #184819 | Available from Addgene |
| Recombinant DNA reagent | p3E_he1a:YFP | Addgene | Addgene plasmid #113879 | |
| Recombinant DNA reagent | pPRISM-Stop-cmlc2-eGFP | *Wierson et al., 2020* | Addgene kit #1000000154 | |

*Appendix 1 Continued*

| Reagent type (species) or resource | Designation | Source or reference | Identifiers | Additional information |
|---|---|---|---|---|
| Recombinant DNA reagent | *pPRISM-QF2-he1.1:YFP-rln3a-HA* | This manuscript | Addgene plasmid #184820 | Available from Addgene |
| Recombinant DNA reagent | *TOPO-nmba* | This manuscript | Addgene plasmid #184821 | Available from Addgene |
| Recombinant DNA reagent | *TOPO-nmbb* | This manuscript | Addgene plasmid #184822 | Available from Addgene |
| Recombinant DNA reagent | *TOPO-nts* | This manuscript | Addgene plasmid #184823 | Available from Addgene |
| Recombinant DNA reagent | *pSPORT-sst1.1* | *Argenton et al., 1999* | N/A | |
| Sequence-based reagent | *gsc2_sense* | IDT | | 5'TAGGTCACCGCACCATCTTCACAG3' |
| Sequence-based reagent | *gsc2_anti-sense* | IDT | | 5'AACCTGTGAAGATGGTGCGGTGA3' |
| Sequence-based reagent | 2 A_QF2_F | IDT | | 5'AAACCCCGGTCCTATGCCACCCA AGCGCAAA3' |
| Sequence-based reagent | 2 A_QF2_R | IDT | | 5'TTAATTACTAGTTTCACTGTTCGTATG TATTAATGTCGGAG3' |
| Sequence-based reagent | *he1.1:YFP_F* | IDT | | 5'TAGTTCTTTAAACTCAACCACTCC AGGCATAGC3' |
| Sequence-based reagent | *he1.1:YFP_R* | IDT | | 5'TCCGCCTCAGAAGCCATAGAGCC CACCGCATC3' |
| Sequence-based reagent | polyA_F | IDT | | 5'TACGAACAGTGAAACTAGTAATTAA GTCTCAGCCAC3' |
| Sequence-based reagent | polyA_R | IDT | | 5'TGGAGTGGTTGAGTTTAAAGAACT AGGAACGCC3' |
| Sequence-based reagent | Col1E_F | IDT | | 5'TGGGCTCTATGGCTTCTGAGGC GGAAAGAAC3' |
| Sequence-based reagent | Col1E_R | IDT | | 5'CTTGGGTGGCATAGGACCGG GGTTTTCTTC3' |
| Sequence-based reagent | *rln3a_5'arm_sense* | IDT | | 5'GCGGTTTCTCGGCTCTCGTAGTGT GTCTGCTGCTGGCTGGAGTAAAGG CGCTGGAC3' |
| Sequence-based reagent | *rln3a_5'arm_anti-sense* | IDT | | 5'GAAGGTCCAGCGCCTTTACTCCAGCCAGCA GCAGACACACTACGAGAGCCGAGAAA3' |
| Sequence-based reagent | *rln3a_3'arm_sense* | IDT | | 5'CGGTTTCGGATGAACTCCCT GCCGCATAATTTGA CTCCATACGAGGGCCCGGCG3' |
| Sequence-based reagent | *rln3a_3'arm_anti-sense* | IDT | | 5'AAGCGCCGGGCCCTCGTATGGAGTC AAATTATGCGGCAGGGAGTTCATCCGAAA3' |
| Sequence-based reagent | *rln3a_sense* | IDT | | 5'TAATACGACTCACTATAGGAGTAAAGG CGCTGGACGCGTTTTAGAGCTAGAAATAGC3' |
| Sequence-based reagent | ugRNA_sense | IDT | | 5'TAATACGACTCACTATAGGGAGGCGTT CGGGCCACAGGTTTTAGAGCTAGAAATAGC3' |
| Sequence-based reagent | common_anti-sense | IDT | | 5'AAAAGCACCGACTCGGTGCCACTTTTT CAAGTTGATAACGGACTAGCCTTATTTTAAC TTGCTATTTCTAGCTCTAAAAC3' |
| Sequence-based reagent | *gsc2_val_F* | IDT | | 5'GTCTGGGGAAAGCGTGTGTT3' |
| Sequence-based reagent | hsp70_R | IDT | | 5'TCAAGTCGCTTCTCTTCGGT3' |
| Sequence-based reagent | *rln3a_val_F* | IDT | | 5'CGCTTTTGTTTCCAGAAAGG3' |

*Appendix 1 Continued on next page*

*Appendix 1 Continued*

| Reagent type (species) or resource | Designation | Source or reference | Identifiers | Additional information |
|---|---|---|---|---|
| Sequence-based reagent | QF2_R | IDT | | 5'CAGACCCGGAGTATCGATGT3' |
| Sequence-based reagent | gsc2_F | IDT | | 5'GTGCAGGACAAGAGGAGCTT3' |
| Sequence-based reagent | gsc2_R | IDT | | 5'GTTTCAATTTAGGTGACACTATAGT CCTCGAAGACTGAAGGGAA3' |
| Sequence-based reagent | rln3a_F | IDT | | 5'CACAGATGAAATCCTGGACTTGT3' |
| Sequence-based reagent | rln3a_R | IDT | | 5'GTTTCAATTTAGGTGACACTATAGCT GAAATGAGAGAGCGAGCA3' |
| Sequence-based reagent | ccka_F | IDT | | 5'TCTGTGTATGTGCCCTGCTG3' |
| Sequence-based reagent | ccka_R | IDT | | 5'GTTTCAATTTAGGTGACACTATA GTGGCCAGTAGTTCGGTTAGG3' |
| Sequence-based reagent | cckb_F | IDT | | 5'GGGGTGTGTGTGTGTGTGAT3' |
| Sequence-based reagent | cckb_R | IDT | | 5'GTTTCAATTTAGGTGACACTAGA GATGAGTTTGGCCAGCAG3' |
| Sequence-based reagent | nmba_F | IDT | | 5'ATGGCTGATGATGGACATTG3' |
| Sequence-based reagent | nmba_R | IDT | | 5'CATCCTGTTGGCCAATTCTT3' |
| Sequence-based reagent | nmbb_F | IDT | | 5'CAGTCCAAGCGTATCCAGGT3' |
| Sequence-based reagent | nmbb_R | IDT | | 5'TCATTTATTGTCTTGAATGTAGCTTT3' |
| Sequence-based reagent | nts_F | IDT | | 5'TTGTGTGTTTTCTCCCTCTTCA3' |
| Sequence-based reagent | nts_R | IDT | | 5'CGGCCGTCTGGATTTATTAG3' |
| Other | Raw data—part 1 | Mendeley Data | http://dx.doi.org/10.17632/tm2bjzjp5g.1 | See Data and Code Availability in Methods |
| Other | Raw data—part 2 | Mendeley Data | http://dx.doi.org/10.17632/mcbdr53ppt.1 | See Data and Code Availability in Methods |
| Other | Raw data—part 3 | Mendeley Data | http://dx.doi.org/10.17632/3vrhjh6xrp.1 | See Data and Code Availability in Methods |
| Other | Raw data—part 4 | Mendeley Data | http://dx.doi.org/10.17632/p9nd6mf7w2.1 | See Data and Code Availability in Methods |
| Other | Raw data—part 5 | Mendeley Data | http://dx.doi.org/10.17632/pmpxtfv2ps.1 | See Data and Code Availability in Methods |
| Other | Raw data—part 6 | Mendeley Data | http://dx.doi.org/10.17632/xwtjpvd885.1 | See Data and Code Availability in Methods |
| Software | *Figure 6* code | Zenodo | http://dx.doi.org/10.5281/zenodo.6412939 | |
| Software | *Figure 7* code | Zenodo | http://dx.doi.org/10.5281/zenodo.6412965 | |
| Software | *Figure 8* code | Zenodo | http://dx.doi.org/10.5281/zenodo.6412969 | |
| Software | *Figure 8—video 1* code | Zenodo | http://dx.doi.org/10.5281/zenodo.6412978 | |

