## [Editor Report · eLife assessment]

This study presents an **important** finding on the anatomical connectivity and functional roles of the previously uncharacterized neuronal populations in the nucleus incertus. The evidence supporting the conclusions is **convincing**, with imaging and manipulations of the genetically targeted populations of neurons. The work presents a significant milestone for future mechanistic studies of the nucleus incertus.

---

## [Referee Report · Reviewer #1 (Public Review)]

Spikol et al. investigate the roles of two distinct populations of neurons in the nucleus incertus (NI). The authors established two new transgenic lines that label gsc2- and rln3a-expressing neurons. They show that the gsc2+ and rln3a+ NI neurons show divergent projection patterns and project to different parts of the interpeduncular nucleus (IPN), which receive inputs from the habenula (Hb). Furthermore, calcium imaging shows that gsc2 neurons are activated by the optogenetic activation of the dorsal Hb-IPN and respond to aversive electric shock stimuli, while rln3a neurons are highly spontaneously active. The ablation of rln3a neurons, but not gsc2 neurons, alters locomotor activity of zebrafish larvae.

The strength of the paper is their genetic approach that enabled the authors to characterize many different features of the two genetically targeted populations in the NI. These two neuronal populations are anatomically closely apposed and would have been indistinguishable without their genetic tools. Their analyses provide valuable information on the diverse anatomical, physiological and behavioral functions of the different NI subtypes. On the other hand, these pieces of evidence are loosely linked with each other to reach a mechanistic understanding of how the NI works in a circuit. For example, the anatomical study revealed the connections from the NI to the IPN, while the optogenetic mapping experiments investigate the other way around, i.e. the connection from the IPN to the NI.

---

## [Referee Report · Reviewer #3 (Public Review)]

This study uses a range of methods to characterize heterogeneous neural populations within the nucleus incertus (NI). The authors focus on two major populations, expressing gsc2 and rln3a, and present convincing evidence that these cells have different patterns of connectivity, calcium activity and effects on behavior. Although the study does not go as far as clarifying the role of NI in any specific neural computation or aspect of behavioral control, the findings will be valuable in support of future endeavors to do so. In particular, the authors have made two beautiful knock-in lines that recapitulate endogenous expression pattern of gsc2 and rln3a which will be a powerful tool to study the roles of the relevant NI cells. Experiments are well done, data are high quality and most claims are well supported. In this revised version, the authors have added additional analysis that has clarified their results and strengthened some of the claims.

Two points of note:

• The data very clearly show different patterns of neurites for gsc2 and rln3a neurons in the IPN and the authors interpret these are being axonal arbors. However, they do not rule out the possibility that some of the processes might be dendritic in nature. Of relevance to this point, they cite a recent study (Petrucco et al. 2023) that confirmed that, as in other species, tegmental neurons in zebrafish extend spatially segregated dendritic as well as axonal arbors into IPN, and the authors speculate that these GABAergic tegmental cells might in fact be part of NI.

• Although the gsc2 and rln3a populations show differences in calcium activity, there is not as clear a dichotomy as stated in the abstract. For example, both populations clearly respond to electric shocks, albeit with different response time courses.

---

## [Referee Report · Reviewer #4 (Public Review)]

Summary:

In the present study, Spikol et al. explore the projection patterns and functional characteristics of two distinct and genetically defined populations in the larval zebrafish Nucleus Incertus (NI), expressing the transcription factor gsc2 or the neuropeptide rln3a. To label in vivo these neurons two transgenic lines were generated by CRISPR/Cas9 mediated Knock-in. These genetic tools allowed the analysis of the projection patterns of these neuronal populations showing that the NI neurons expressing gsc2 and rln3a exhibit markedly different projection patterns, targeting separate subregions within the midbrain interpeduncular nucleus (IPN).

Functional imaging and behavioral analysis revealed that while gsc2 neurons respond to electric shock stimuli, rln3a neurons show high spontaneous activity and play a role in regulating locomotor activity.

Strengths:

The paper relies on a series of rigorous experimental approaches including molecular genetic, neuroanatomical, functional and behavioral analysis. The resources generated including the two knock-in transgenic reporter lines will be of great value for the zebrafish neurobiology community as well as inspire further studies of the NI in other model systems.

Weaknesses:

Technical weaknesses present in the first version of the manuscript have largely been addressed in the present revision.

---

## [Author Response]

The following is the authors’ response to the original reviews.

We are thankful to the reviewers and the editor for their detailed feedback, insightful suggestions, and thoughtful assessment of our work. Our point-by-point responses to the comments and suggestions are below.

The revised manuscript has taken into account all the comments of the three reviewers. Modifications include corrections to errors in spelling and unit notation, additional quantification, improvements to the clarity of the language in some places, as well as additional detail in the descriptions of the methods, and revisions to the figures and figure legends.

We have also undertaken additional analyses and added materials in response to reviewer suggestions. In brief:

In response to a suggestion from Reviewer #1, we added Figure 6-1 to show examples of the calcium traces of individual fish and individual ROIs from the condensed data in Figure 6.We revised Figure 7 as follows:

We added an analysis of the duration of the response to shock to address comments from Reviewers #2 and #3.In response to Reviewer #3, we added histograms showing the distribution of the amplitudes of the calcium signals in the gsc2 and rln3a neurons to show, without relying on the detection of peaks in the calcium trace, that the rln3a neurons have more oscillations in activity.

We added Figure 8-2 in response to the suggestion from Reviewer #3 to analyze turning behavior in larvae with ablated rln3a neurons.

To address Reviewer #2’s suggestion to show how the ablated transgenic animals compare to the non-ablated transgenic animals of the same genotype, we have added this analysis as Figure 8-3.

A detailed point-by-point is as follows:

The reviewers agree that the study of Spikol et al is important, with novel findings and exciting genetic tools for targeting cell types in the nucleus incertus. The conclusions are overall solid. Results could nonetheless be strengthened by performing few additional optogenetic experiments and by consolidating the analysis of calcium imaging and behavioral recordings as summarized below.(1) Light pulses used for optogenetic-mediated connectivity mapping were very long (5s), which could lead to non specific activation of numerous population of neurons than the targeted ones. To confirm their results, the authors should repeat their experiments with brief 5-50ms (500ms maximum) -long light pulses for stimulation.

As the activity of the gsc2 neurons is already increased by 1.8 fold (± 0.28) within the first frame that the laser is activated (duration ~200 msec), it is unlikely that that the observed response is due to non-specific activation induced by the long light pulse.

(2) In terms of analysis, the authors should improve :

a) The detection of calcium events in the "calcium trace" showing the change in fluorescence over time by detecting the sharp increase in the signal when intracellular calcium rises;

We have added an additional analysis to Figure 7 that does not rely on detection of calcium peaks. See response to Reviewer #3.

b) The detection of bouts in the behavioral recordings by measuring when the tail beat starts and ends, thereby distinguishing the active swimming during bouts from the immobility observed between bouts.

Our recordings capture the entire arena that the larva can explore in the experiment and therefore lack the spatial resolution to capture and analyze the tail beat. Rather, we measured the frequency and length of phases of movement in which the larva shows no more than 1 second of immobility. To avoid confusion with studies that measure bouts from the onset of tail movement, we removed this term from the manuscript and refer to activity as phases of movement.

(3) The reviewers also ask for more precisions in the characterization of the newly-generated knock-in lines and the corresponding anatomy as explained in their detailed reports.

Please refer to the point-by-point request for additional details that have now been added to the manuscript.

**Reviewer #1 (Recommendations For The Authors):**
The conclusions of this paper are mostly well supported by data, but some technical aspects, especially about calcium imaging and data analysis, need to be clarified.(1) Both the endogenous gsc2 mRNA expression and Tg(gsc2:QF2) transgenic expression are observed in a neuronal population in the NI, but also in a more sparsely distributed population of neurons located more anteriorly (for example, Fig. 2B, Fig. 5A). The latter population is not mentioned in the text. It would be necessary to clarify whether or not this anterior population is also considered as the NI, and whether this population was included for the analysis of the projection patterns and ablation experiments.

The sparsely distributed neurons had been mentioned in the Results, line 134, but we have now added more detail. In line 328, we have clarified that: “As the sparsely distributed anterior group of gsc2 neurons (Fig. 2B, C) are anatomically distinct from the main cluster and not within the nucleus incertus proper, they were excluded from subsequent analyses.”

(2) Both Tg(gsc2:QF2) and Tg(rln3a:QF2) transgenic lines have the QF genes inserted in the coding region of the targeted genes. This probably leads to knock out of the gene in the targeted allele. Can the authors mention whether or not the endogenous expression of gsc2 and rln3a was affected in the transgenic larvae? Is it possible that the results they obtained using these transgenic lines are affected by the (heterozygous or homozygous) mutation of the targeted genes?

Figure 8-1 includes in situ hybridization for gsc2 and rln3a in heterozygous Tg(gsc2:QF2)c721; Tg(QUAS:GFP)c578 and Tg(rln3a:QF2; he1.1:YFP)c836; Tg(QUAS:GFP)c578 transgenic larvae.

The expression of gsc2 is unaffected in Tg(gsc2:QF2)c721; Tg(QUAS:GFP)c578 heterozygotes

(Fig. 8-1A), whereas the expression of rln3a is reduced in Tg(rln3a:QF2; he1.1:YFP)c836;Tg(QUAS:GFP)c578 heterozygous larvae (Fig. 8-1D), as mentioned in the legend for Figure 8-1. We confirmed these findings by comparing endogenous gene expression between transgenic and non-transgenic siblings that were processed for RNA in situ hybridization in the same tube.

The behavioral results we obtained are not due to rln3a heterozygosity because comparisons were made with sibling larvae that are also heterozygous for Tg(rln3a:QF2; he1.1:YFP)c836; Tg(QUAS:GFP)c578, as stated in the Figure 8 legend.

(3) Optogenetic activation and simultaneous calcium imaging is elegantly designed using the combination of the orthogonal Gal4/UAS and QF2/QUAS systems (Fig. 6). However, I have some concerns about the analysis of calcium responses from a technical point of view. Their definition of ΔF/F in this manuscript is described as (F-Fmin)/(Fmax-Fmin) (see line 1406). This is confusing because it is different from the conventional definition of ΔF/F, which is F-F0/F0, where F0 is a baseline GCaMP fluorescence. Their way of calculating the ΔF/F is inappropriate for measuring the change in fluorescence relative to the baseline signal because it rather normalizes the amplitude of the responses across different ROIs. The same argument applies to the analyses done for Fig. 7.

We have taken a careful look at our analyses and replotted the data using F-F0/F0. However, this only changes Y-axis values and does not change the shape of the calcium trace or the change in signal upon stimulation. Both metrics (F-F0/F0 and (F-Fmin)/(Fmax-Fmin)) adjust the fluorescence values of each ROI to its own baseline.

(4) The %ΔF/F plots shown in Fig.6 are highly condensed showing the average of different ROIs (cells) within one fish and then the average of multiple fish. It would be helpful to see example calcium traces of individual ROIs and individual fish to know the variability across ROIs and fish. Also, It would be helpful to know how much laser power (561 nm laser) was used to photostimulate ReaChR.

Laser power (5%) was added to the section titled Calcium Signaling in Methods.

In Figure 6, shading in the %ΔF/F plots (D, D’, E, E’, F, F’, G, G’, H, H’) represents the variability across ROIs, and the dot plots (D’’, E’’, F’’, G’’, H’’) show the variability across fish (where each data point represents an individual fish). We have now also added Figure 6-1 with examples of calcium traces from individual fish and individual ROIs.

(5) Some calcium traces presented in Fig. 6 (Fig. 6D, D', F, H, H') show discontinuous fluctuations at the onset and offset of the photostimulation period. Is this caused by some artifacts introduced by switching the settings for the photostimulation? The authors should mention if there are some alternative explanations for this discontinuity.

As noted by the reviewer, this artifact does result from switching the settings for photostimulation, which we mention in the legend for Figure 6.

(6) In the introduction, they mention that the griseum centrale is a presumed analogue of the NI (lines 74-75). It would be helpful for the readers to better understand the brain anatomy if the authors could discuss whether or not their findings on the gsc2 and rln3a NI neurons support this idea.

Our findings on the gsc2 and rln3a neurons support the idea that the griseum centrale of fish is the analogue of the mammalian NI. We have now edited the text in the third paragraph of the discussion, line 1271, to make this point more clearly: “By labeling with QUAS-driven fluorescent reporters, we determined that the anatomical location, neurotransmitter phenotype, and hodological properties of gsc2 and rln3a neurons are consistent with NI identity, supporting the assertion that the griseum centrale of fish is analogous to the mammalian NI. Both groups of neurons are GABAergic, reside on the floor of the fourth ventricle and project to the interpeduncular nucleus.”

**Reviewer #2 (Recommendations For The Authors):**
Major comments:(1) Throughout the figures a need for more precision and reference in the anatomical evidence:

Specify how many planes over which height were projected for each Z-projection in Figure1,2,3, ....

We added this information to the last paragraph of the section titled Confocal Imaging within the Materials and Methods.

Provide the rhombomere numbers, deliminate the ventricles & always indicate on the panel the orientation (Rostral Caudal, Left Right or Ventral Dorsal) for Figure 1 panels D-F , Figure 2-1B-G, Figure 2-2A-C in the adult brain, Figure 3.

We annotated Figures 2-1 and 2-2 as suggested. We also indicated the orientation (anterior to the top or anterior to the left) in all figure legends. For additional context on the position of gsc2 and rln3a neurons within the larval brain, refer to Fig. 1A-C’, Fig. 1-2A, Fig. 2, Fig. 4 and Fig. 5.

Add close up when necessary: Figure 2-2A-C, specify in the text & in the figure where are the axon bundles from the gsc2+ neurons in the adult brain- seems interesting and is not commented on?

We added a note to the legend of Figure 2-2: Arrowheads in B and B’ indicate mApple labeling of gsc2 neuronal projections to the hypothalamus. We also refer to Fig 2-2B, B’ in the Results section titled Distinct Projection Patterns of gsc2 and rln3a neurons.

keep the same color for one transgene within one figure: example, glutamatergic neurons should always be the same color in A,B,C - it is confusing as it is.

We have followed the reviewer’s suggestion and made the color scheme consistent in Figure 3.

Movies: add the labels (which transgenic lines in which color, orientation & anatomical boundaries for NI, PAG, any other critical region that receives their projections and the brain ventricle boundaries) on the anatomical movies in supplemental (ex Movie 4-1 for gsc2 neurons and 4-2 for rln3 neurons: add cerebellum, IPN, raphe, diencephalon, and rostral and caudal hypothalamus, medulla for 4-1 as well as lateral hypothalamus and optic tectum for 42); add the ablated region when necessary.

We added more detail to the movie legends. Please refer to Figure 4 for additional anatomical details.

for highlighting projections from NI neurons and distinguish them from the PAG neurons, the authors elegantly used 2 Photon ablation of one versus the other cluster: this method is valid but we need more resolution that the Z stacks added in supplemental by performing substraction of before and after maps.

We are not sure what the author meant by subtraction as there are no before and after images in this experiment. Larvae underwent ablation of cell bodies and were imaged one day later in comparison to unablated larvae.

In particular, it is not clear to me if both PAG and NI rln3a neurons project to medulla - can the authors specify this point & the comparison between intact & PAG vs NI ablation maps? The authors should resolve better the projections to all targeted regions of NI gsc2 neurons and differentiate them from other PAG gsc2 neurons, same for rln3a neurons.

We have clarified this point on line 549.

Make sure to mention in the result section the duration between ablation & observation that is key for the axons to degrade.

We always assessed degeneration of neuronal processes at 1-day post-ablation.

(“2) calcium imaging experiments:

a) with optogenetic connectivity mapping:

the authors combine an impressive diverse set of optogenetic actuators & sensors by taking advantage of the QUAS/QF2 and UAS/GAL4 systems to test connectivity from Hb-IPN onto gsc2 and rln3 neurons.The experiments are convincing but the choice of the duration of the stimulation (5s) is not adequate to test for direct connectivity: the authors should make sure that response in gsc2 neurons is observed with short duration (50ms-1s max).

As noted above:

“As the activity of the gsc2 neurons is already increased by 1.8 fold (± 0.28) within the first frame that the laser is activated (duration ~200 msec), it is unlikely that that the observed response is due to non-specific activation induced by the long light pulse.”

note: Specify that the gsc2 neurons tested are in NI.

We have edited the text accordingly in the Results section titled Afferent input to the NI from the dHb-IPN pathway.

b) for the response to shock:in the example shown for rln3 neurons, the activity differs before and after the shock with long phases of inhibition that were not seen before. Is it representative? the authors should carefully stare at their data & make sure there is no difference in activity patterns after shock versus before.

We reexamined the responses for each of the rln3a neurons individually and confirmed that, although oscillations in activity are frequent, the apparent inhibition (excursions below baseline) are an idiosyncratic feature of the particular example shown.

(3) motor activity assay:a) there seems to be a misconception in the use of the word "bout" to estimate in panels H and I bout distance and duration and the analysis should be performed with the criterion used by all in the motor field:As we know now well based on the work of many labs on larval zebrafish (Orger, Baier, Engert, Wyart, Burgess, Portugues, Bianco, Scott, ...), a bout is defined as a discrete locomotor event corresponding to a distance swam of typically 1-6mm, bout duration is typically 200ms and larvae exhibit a bout every s or so during exploration (see Mirat et al Frontiers 2013; Marques et al Current Biology 2018; Rajan et al. Cell Reports 2022).Since the larval zebrafish has a low Reynolds number, it does not show much glide and its movement corresponds widely to the active phase of the tail beats.Instead of detecting the active (moving) frames as bouts, the authors however estimate these values quite off that indicate an error of calibration in the detection of a movement: a bout cannot last for 5-10s, nor can the fish swim for more than 1 cm per bout (in the definition of the authors, bout last for 5-10 s, and bout correspond to 10 cm as 50 cm is covered in 5 bouts).The authors should therefore distinguish the active (moving) from inactive (immobile) phase of the behavior to define bouts & analyze the corresponding distance travelled and duration of active swimming. They would also benefit from calculating the % of time spent swimming in order to test whether the fish with ablated rln3 neurons change the fraction of the time spent swimming.

As noted above:

Our recordings capture the entire arena that the larva can explore in the experiment and therefore lack the spatial resolution to capture and analyze the tail beat. Rather, we measured the frequency and length of phases of movement in which the larva shows no more than 1 second of immobility. To avoid confusion with studies that measure bouts from the onset of tail movement, we removed this term from the manuscript and refer to activity as phases of movement.

Note that a duration in seconds is not a length and that the corresponding symbol for seconds in a scientific publication is "s" and not "sec".

We have corrected this.

b) controls in these experiments are key as many clutches differ in their spontaneous exploration and there is a lot of variation for 2 min long recordings (baseline is 115s). The authors specify that the control unablated are a mix of siblings; they should show us how the ablated transgenic animals compare to the non ablated transgenic animals of the same clutch.

The unablated Tg(gsc2:QF2)c721; Tg(QUAS:GFP)c578 and Tg(rln3a:QF2, he1.1:YFP)c836;Tg(QUAS:GFP)c578 larvae in the control group are siblings of ablated larvae. We repeated the analyses using either the Tg(gsc2:QF2)c721; Tg(QUAS:GFP)c578 or Tg(rln3a:QF2, he1.1:YFP)c836; Tg(QUAS:GFP)c578 larvae only as controls and added the results in Figure 8-3. Although the statistical power is slightly reduced due to a smaller number of samples in the control group, the conclusions are the same, as the behavior of Tg(gsc2:QF2)c721; Tg(QUAS:GFP)c578 and Tg(rln3a:QF2, he1.1:YFP)c836; Tg(QUAS:GFP)c578 unablated larvae is indistinguishable.

Minor comments:(1) Anatomy :

Add precision in the anatomy in Figure 1:Improve contrast for cckb.

The contrast is determined by the signal to background ratio from the fluorescence in situ hybridization. Increasing the brightness would increase both the signal and the background, as any modification must be applied to the whole image.

since the number of neurons seems low in each category, could you quantify the number of rln3+, nmbb+, gsc2+, cckb+ neurons in NI?

Quantification of neuronal numbers has been added to the first Results section titled Identification of gsc2 neurons in the Nucleus Incertus, lines 219-224.

note: indicate duration for the integral of the DF/F in s and not in frames.

We have added this in the legends for Figures 6 and 7 and in Materials and Methods.

(2) Genetic tools:To generate a driver line for the rln3+ neurons using the Q system, the authors used the promoter for the hatching gland in order to drive expression in a structure outside of the nervous system that turns on early and transiently during development: this is a very elegant approach that should be used by many more researchers.If the her1 construct was integrate together with the QF2 in the first exon of the rln3 locus as shown in Figure 2, the construct should not be listed with a ";" instead of a "," behind rln3a:QF2 in the transgene name. Please edit the transgene name accordingly.

We have edited the text accordingly.

(3) Typos:GABAergic neurons is misspelled twice in Figure 3.

Thank you for catching this. We have corrected the misspellings.

**Reviewer #3 (Recommendations For The Authors):**
More analysis should be done to better characterize the calcium activity of gsc2 and rln3a populations. Specifically:Spontaneous activity is estimated by finding peaks in the time-series data, but the example in Fig7 raises concerns about this process: Two peaks for the gsc2 cell are identified while numerous other peaks of apparently similar SNR are not detected. Moreover, the inset images suggest GCaMP7a expression might be weaker in the gsc2 transgenic and as such, differences in peak count might be related to the SNR of the recordings rather than underlying activity. Overall, the process for estimating spontaneous activity should be more rigorous.

To not solely rely on the identification of peaks in the calcium traces, we also plotted histograms of the amplitudes of the calcium signals for the rln3a and gsc2 neurons. The histograms show that the amplitudes of the rln3a calcium signals frequently occur at small and large values (suggesting large fluctuations in activity), whereas the amplitudes of the gsc2 calcium signals occur most frequently at median values. We added this analysis to a revised Figure 7.

Interestingly, there are a number of large negative excursions in the calcium data for the rln3a cell - what is the authors' interpretation of these? Could it be that presynaptic inhibition via GABA-B receptors in dIPN might influence dIPN-innervating rln3a neurons?

As noted above:

We reexamined the responses for each of the rln3a neurons individually and confirmed that, although oscillations in activity are frequent, the apparent inhibition (excursions below baseline) are an idiosyncratic feature of the particular example shown.

Regarding shock-evoked activity, the authors state "rln3a neurons showed ... little response to shock", yet the immediate response after shock appears very similar in gsc2 vs rln3a cells (approx 30 units on the dF/F scale). The subsequent time-course of the response is what appears to distinguish gsc2 versus rln3a; it might thus be useful to separately quantify the amplitude and decay time constant of the shock evoked response for the two populations.

The reviewer is correct that the difference between the gsc2 and rln3a neurons in the response to shock is dependent on the duration of time post-shock that is analyzed. Thus, the more relevant feature is the length of the response rather than the size. To reflect this, we compared the average length of responses for the gsc2 and rln3a neurons. We have now added this analysis to Figure 7 and updated the text accordingly.

The difference in spontaneous locomotor behavior is interesting and the example tracking data suggests there might also be differences in turn angle distribution and/or turn chain length following rln3 NI ablations. I would recommend the authors consider exploring this.

Thank you for this suggestion. We wrote additional code to quantify turning behavior and found that larvae with rln3a NI neurons ablated do indeed have a statistically significant increase in turning compared to other groups. We now show this analysis as Figure 8-2 and we added an explanation of the quantification of turning behavior to the Methods section titled Locomotor assay.

I didn't follow the reasoning in the discussion that activity of rln3a cells
may control transitions between phases of behavioral activity and .
nts (at least those that are detected) in Fig7 occur with an average interval exceeding 30 s, yet swim bouts occur at a frequency around 1 Hz. The authors should clarify their hypothesis about how these disparate timescales might be connected.

As noted above:

Our recordings capture the entire arena that the larva can explore in the experiment and therefore lack the spatial resolution to capture and analyze the tail beat. Rather, we measure the frequency and length of phases of movement in which the larva shows no more than 1 second of immobility. To avoid confusion with studies that measure bouts from the onset of tail movement, we removed this term from the manuscript and refer to activity as phases of movement.

Fig2-2:
Images are ordered from (A, B, C) anterior to (A', B', C') posterior.
Its not clear what this means and images appear to be in sequence A, A', B, B'.... please clarify and consider including a cartoon of the brain in sagittal view showing location of sections indicated.

We clarified the text in the Figure 2-2 legend and added a drawing of the brain showing the location of the sections.

In Fig7, why are 300 frames analyzed pre/post shock? Even for gsc2, the response appears complete in ~100 frames.

Reviewer #2 also pointed out that the difference between the gsc2 and rln3a neurons in the response to shock is dependent on the duration of time post-shock that is analyzed. Thus, the more relevant feature is the length of the response rather than the size. To reflect this, we compared the average length of response for the gsc2 and rln3a neurons and modified the text and Figure as described above.

What are the large negative excursions in the calcium signal in the rln3a data (Fig7E)?

See response to Reviewer # 2, repeated below:

We looked through each of the responses of individual rln3a neuron and confirmed that, although oscillations in activity are frequent among the rln3a neurons, the apparent inhibition (excursions below baseline) are an idiosyncratic feature of the particular example shown.

There are several large and apparently perfectly straight lines in the fish tracking examples (Fig8) suggestive of tracking errors (ie. where the tracked centroid instantaneously jumps across the camera frame). Please investigate these and include analysis of the distribution of swim velocities to support the validity of the tracking data.

The reason for this is indeed imperfect tracking resulting in frames in which the tracker does not detect the larva. The result is that the larva appears to move 1 cm or more in a single frame. However, analysis of the distribution of distances across all frames shows that these events (movement of 1 cm or more in a single frame) are rare (less than 0.04%), and there are no systematic differences that would explain the differences in locomotor behavior presented in Fig. 8. A summary of the data is as follows:

Controls: 0.0249% of distances 1 cm or greater gsc2 neurons ablated: 0.0302% of distances 1 cm or greater rln3a NI neurons ablated: 0.0287% of distances 1 cm or greater rln3a PAG neurons ablated: 0.0241% of distance 1 cm or greater

Insufficient detail is provided in the methods about how swim bouts are detected (and their durations extracted) from the centroids tracking data. Please expand detail in this section.

We added an explanation to the Methods section titled Locomotor assay.